# Growth in early infancy drives optimal brain functional connectivity which predicts cognitive flexibility in later childhood

Chiara Bulgarelli[1,2]*, Anna Blasi[2], Samantha McCann[3,4], Bosiljka Milosavljevic[5,6], Giulia Ghillia[3,7], Ebrima Mbye[4], Ebou Touray[4], Tijan Fadera[4], Lena Acolatse[4,8], Sophie E Moore[3,4], Sarah Lloyd-Fox[5], Clare E Elwell[2], Adam T Eggebrecht[9], The BRIGHT Study Team

[1]Centre for Brain and Cognitive Development, Birkbeck, University of London, London, United Kingdom; [2]Department of Medical Physics and Biomedical Engineering, University College London, London, United Kingdom; [3]Department of Women and Children's Health, King's College London, London, United Kingdom; [4]Medical Research Council Unit The Gambia at the London School of Hygiene and Tropical Medicine, Banjul, Gambia; [5]Department of Psychology, University of Cambridge, Cambridge, United Kingdom; [6]School of Biological and Experimental Psychology, Queen Mary University of London, London, United Kingdom; [7]Department of Women and Children's Health, University of Liverpool, Liverpool, United Kingdom; [8]Nutrition Innovation Centre for Food and Health, School of Biomedical Sciences, Ulster University, Coleraine, Ireland; [9]Mallinckrodt Institute of Radiology, Washington University School of Medicine in St. Louis, St. Louis, United States

*For correspondence:
c.bulgarelli@bbk.ac.uk

Group author details:
The BRIGHT Study Team See page 14

## eLife Assessment

This **important** study details changes in the brain functional connectivity in a longitudinal cohort of Gambian children assessed outside a lab setup with functional near-infrared spectroscopy (fNIRS) from age 5 to 24 months, in relation to early physical growth and cognitive flexibility capacities at preschool age. Evidence supporting conclusions on the evolution of brain connectivity is **convincing** and highlights a different trajectory compared with populations from high-income countries. However, analyses linking connectivity trajectories with early adverse conditions such as undernutrition and later cognitive development are only partially supported due to insufficient longitudinal data and statistical power. This study will be of significant interest to neuroscientists, psychologists and neuroimaging researchers working on infant development in relation to environmental factors.

**Abstract** Functional brain network organisation, measured by functional connectivity (FC), reflects key neurodevelopmental processes for healthy development. Early exposure to adversity, for example undernutrition, affects neurodevelopment, observable via disrupted FC, and leads to poorer outcomes from preschool age onwards. We assessed longitudinally the impact of early growth trajectories on developmental FC in a rural Gambian population from age 5–24 months. To investigate how these early trajectories relate to later childhood outcomes, we assessed cognitive flexibility at 3–5 years. We observed that early physical growth before the fifth month of life drove

optimal developmental trajectories of FC that in turn predicted cognitive flexibility at pre-school age. In contrast to previously studied developmental populations, this Gambian sample exhibited long-range interhemispheric FC that decreased with age. Our results highlight the measurable effects that poor growth in early infancy has on brain development and the possible subsequent impact on pre-school age cognitive development, underscoring the need for early life interventions throughout global settings of adversity.

## Introduction

The first 1000 days of life are of paramount importance for human brain (*Kang et al., 2011*) and body development (*Black et al., 2017*; *Goyal and Raichle, 2013*). Early adversity negatively impacts infant development, which leads to long-term consequences from delayed milestones in childhood development to lowered productivity at the individual and societal levels (*Nelson, 2000*; *Rod et al., 2020*; *Kim et al., 2013*). Therefore, assessing trajectories of brain development in infants exposed to early adversity is essential to understand and mitigate effects on brain development and subsequent outcomes at school age and beyond (*Kuzawa et al., 2014*; *Thompson and Nelson, 2001*).

Early undernutrition engenders detrimental effects on a range of cognitive skills (*Kumar Kesari Yassin et al., 2010*), with long-lasting effects through adulthood (*Victora et al., 2008*). Interventions using therapeutic feeding regimens and supplements have been shown to have a strong causal impact on childhood undernutrition, brain development, and subsequent neurodevelopmental outcomes (*Grantham-McGregor et al., 1991*; *Cusick and Georgieff, 2012*; *McKay et al., 1978*). Despite numerous attempts of interventions, global rates of undernutrition remain high (*Moore, 2020*), with infants in low- and middle-income countries (LMICs) at greatest risk (*Nabwera et al., 2017*). Up to one-third of infants in LMICs are at risk of not meeting standard milestones of social, motor, linguistic, and cognitive development by pre-school age (*McCoy et al., 2016*; *Martorell, 1999*), which can lead to lifelong consequences that impact social and economic development at individual, national, and global scales (*Hoddinott et al., 2008*). Thus, it is essential to study neurodevelopment throughout the first 2 years in regions with high rates of undernutrition in order to optimize early interventions. To identify effective early interventions, we must first identify the mechanisms that link undernutrition to later cognitive outcomes. Effects of undernutrition on the developing brain may manifest through alterations in function brain networks (*Black, 2018*), as shown in school-age children exposed to chronic poverty (*Sripada et al., 2014*). However, our understanding of the relationships between early undernutrition, brain connectivity, and later cognitive outcomes is still very limited.

Functional brain network organization is commonly assessed with resting-state functional connectivity (FC) via temporal correlations in brain activity as measured with functional magnetic resonance imaging (fMRI; *Biswal et al., 1995*; *Gao et al., 2009*). Based on data from fMRI, current models hypothesize that FC patterns mature throughout early development (*Gao et al., 2009*; *Gao et al., 2015*; *Fransson et al., 2009*; *Fransson et al., 2007*; *De Asis-Cruz et al., 2015*), where in typically developing brains, adult-like networks emerge over the first years of life as long-range functional connections between pre-frontal, parietal, temporal, and occipital regions become stronger and more selective (*Smyser et al., 2011*; *Damaraju et al., 2014*; *Gao et al., 2017*; *Bulgarelli et al., 2020b*). This maturation in FC has been shown to be related to the cascading maturation of myelination and synaptogenesis (*Betzel et al., 2014*; *Honey et al., 2009*) - fundamental processes for healthy brain development (*Dubois et al., 2014*). Therefore, disrupted patterns of FC may reflect disrupted anatomical maturation of brain circuits and systems. For example, preterm infants with severe diffuse white matter impairment exhibited lower levels of FC in executive function networks compared with preterm infants with no or mild white matter impairment (*He and Parikh, 2015*). Importantly, normative developmental patterns may be disrupted and even reversed in clinical conditions that impact development; for example increased short-range and reduced long-range FC have been observed in preterm infants (*Smyser et al., 2010*) and in children with autism spectrum disorder (*Khan et al., 2013*; *Dajani and Uddin, 2016*).

While widely used neuroimaging modalities such as fMRI can elucidate typical and atypical developmental trajectories, they are often poorly suited for lower resource neuroimaging contexts due to limited portability, need for specialised facilities, and the high cost associated with these methods (*Smyser et al., 2011*; *Smyser et al., 2010*; *Shen et al., 2021*). To address the need for neuroimaging

studies in lower resource environments including (but not limited to) in LMICs, researchers have increasingly turned to more portable tools, including electroencephalography (EEG; *Jensen et al., 2019*), functional near-infrared spectroscopy (fNIRS; *Lloyd-Fox et al., 2017*; *Blasi et al., 2019*; *Lloyd-Fox et al., 2019*), and diffuse correlation spectroscopy (DCS; *Roberts et al., 2017*). These tools enable the creation of mobile neuroimaging laboratories that can be deployed virtually anywhere, eliminating practical constraints imposed by costly and immobile methods. Indeed, research based in The Gambia, Guinea-Bissau, Bangladesh, and Ivory Coast have established feasibility of these tools to perform brain imaging studies outside of a specialised lab environment, with recent studies reporting altered cortical physiology related to early adversity and undernutrition (*Jensen et al., 2019*; *Lloyd-Fox et al., 2019*; *Roberts et al., 2017*; *Jasińska and Guei, 2018*). For example, using EEG, Xie and colleagues showed that in 2–3 year old Bangladeshi children, growth faltering was associated with FC in the theta and beta frequency bands, which was negatively related to children's IQ score at 4 years (*Xie et al., 2019*). With regard to fNIRS, it has a better spatial resolution and anatomical specificity than EEG, thus providing more precise and reliable localisation of brain networks (*Eggebrecht et al., 2014*; *Ferradal et al., 2016*). Moreover, fNIRS facilitates testing in awake and engaged infants, making the results more comparable with adult FC studies that are performed on awake participants (*Mitra et al., 2017*). Given these unique strengths, fNIRS has been used to study FC in longitudinal studies in awake infants in high-resource settings (*Bulgarelli et al., 2020b*), and the portability and the low cost of this equipment has fostered implementation outside conventional labs and in LMICs (*Blasi et al., 2019*). Because fNIRS overcomes logistical (MRI) and resolution (EEG) challenges of other common modalities, we chose it for this study to assess early life brain health of children in LMIC exposed to early severe undernutrition.

The goal of the study was to investigate early physical growth in infancy, developmental trajectories of brain FC across the first 2 years of life, and cognitive outcome at school age in a longitudinal cohort of infants and children from rural Gambia, an environment with high rates of maternal and child undernutrition. Specifically, we aimed to: (i) investigate whether differences in physical growth through the first 2 years of life are related to FC at 24 months, and (ii) investigate if early FC has an impact on cognitive outcome at pre-school age in these children. Data were collected from N=204 children as part of the Brain Imaging for Global Health project (BRIGHT; globalfnirs.org/the-bright-project), a longitudinal study examining infant development from term birth to 24 months of age. We acquired

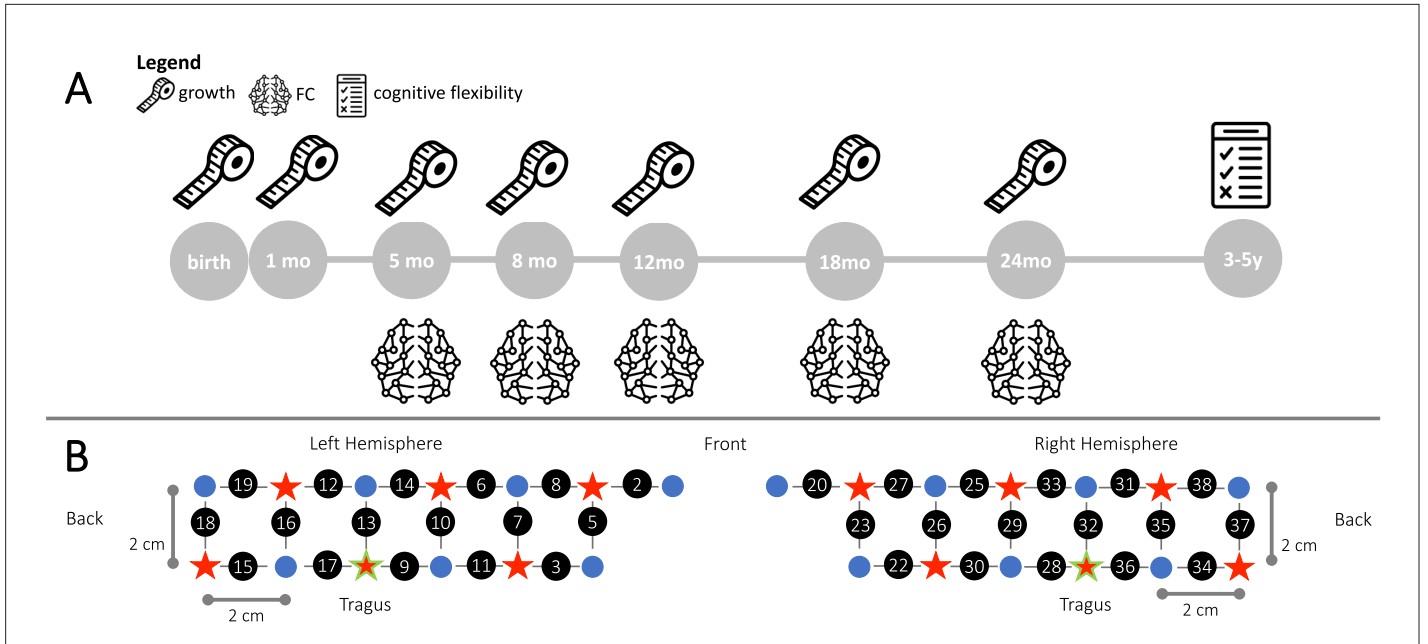

**Figure 1.** Experimental design. (**A**) Measures taken in the BRIGHT project used in this work. The measuring tape represents anthropometric measures, the brain represents fNIRS FC and the test represents the cognitive flexibility assessment. (**B**) Schematic representation of the spatial layout of the fNIRS array. Sources are marked with red stars, detectors are marked with blue circles, channels are marked with grey lines and numbered with black circles. The channels/optodes used as a reference for the tragus are highlighted in green.

longitudinal measurements of brain FC in awake Gambian infant participants at five age points (5, 8, 12, 18, and 24 months) using fNIRS as the infants watched calming videos. We also acquired anthropometric measures (i.e. weight and length) as indicators of nutritional status at birth, at 7-to-14 days, 1 month of age, and at all five fNIRS data acquisition time points. Additionally, because developmental delays interfere with everyday functioning starting within the pre-school age range (*McCoy et al., 2016*), we assessed cognitive flexibility, a core executive function (*Dajani and Uddin, 2015*), at ages 3 and 5 years. We hypothesized that (i) long-range FC would increase and short-range FC would decrease throughout the first 2 years of life, (ii) that early positive physical growth would significantly predict long-range FC, and that (iii) long-range FC strength would predict cognitive flexibility performance in pre-schoolers. Our results provide novel insights into the developmental trajectory of the functional organization of the developing brain, its relationship with nutritional adversity exposure and status, and the subsequent effects on preschool-age cognitive outcomes.

## Results

### Gambian infants exhibit robust FC development throughout first 2 years

The repeated assessments of brain FC throughout the first 2 years of life allowed us to analyse developmental trajectories of the maturation of early brain networks and how they relate to measures of infants' growth as well as to cognitive flexibility at preschool age (*Figure 1A*). To estimate how FC changed between 5 and 24 months of age in Gambian children, we performed linear mixed models (LMM) using data from the 132 infants with valid data for at least two time points. To increase statistical power while minimizing errors due to variability in head anatomy and cap placement, we divided the array into six sections and calculated functional connectivity (FC) between all possible 21 interhemispheric homotopic, intrahemispheric within section, fronto-posterior, and crossed connections (*Appendix 1—figure 1*). Results on the Fisher-z transformed correlation coefficients (z-RHO) of the oxygenated haemoglobin ($HbO_2$) showed that frontal interhemispheric homotopic FC decreased with age ($F=11.03$, $p<0.001$), and that left ($F=5.8$, $p<0.001$) and right fronto-middle ($F=4.86$, $p<0.001$) and right fronto-posterior ($F=5.52$, $p<0.001$) FC increased with age (*Figure 2*). Results on the z-RHO scores of the deoxygenated haemoglobin (HHb) largely agreed with the results on $HbO_2$ (*Table 1* and *Figure 2*). Frontal interhemispheric FC significantly decreased with age, showing positive correlations at 5 months and negative values at 24 months (*Figure 2B and C*). The FC between fronto-middle and frontal-posterior that significantly increased with age showed strong anticorrelation values at early ages that gradually decreased but were still negative at 24 months (*Figure 2B and C*). Notably, results from LMMs performed on data pre-processed without the global signal regression (GSR) (see results in *Appendix 1—table 1* and *Figure 2*) confirmed a decreasing trajectory of the frontal interhemispheric homotopic FC and an increasing trajectory of bilateral short-range FC with age.

### Early growth trajectories predict functional connectivity at 24 months

To investigate the impact of early nutritional status on FC at 24 months, we used multiple regression with the infant growth trajectory (delta weight for length z-score between all time points, ΔWLZ) and FC at 24 months, adjusting for WLZ at birth or head-circumference z-score (HCZ) at 7/14 days. To maximise power, we considered only those FC that showed a statistically significant change with age. The ΔWLZ between birth/1 month and older ages positively predicted frontal interhemispheric homotopic FC at 24 months (ΔWLZ birth with older ages all $p<0.02$ FDR corrected, ΔWLZ 1 month with older ages all $p<0.03$, see *Appendix 1—figure 3* for some examples of scatterplots) and negatively predicted left and right fronto-middle FC at 24 months ($p<0.04$). The ΔWLZ between birth and 1 month positively predicted right fronto-posterior FC at 24 months ($p=0.006$, FDR corrected) while ΔWLZ between 1 month and older ages negatively predicted right fronto-posterior FC at 24 months (all $p<0.01$, FDR corrected). Interestingly, ΔWLZ between 5/8/12/18 months and older ages did not show statistically significant impacts on any of the FC assessed (*Table 2* and *Figure 3A*).

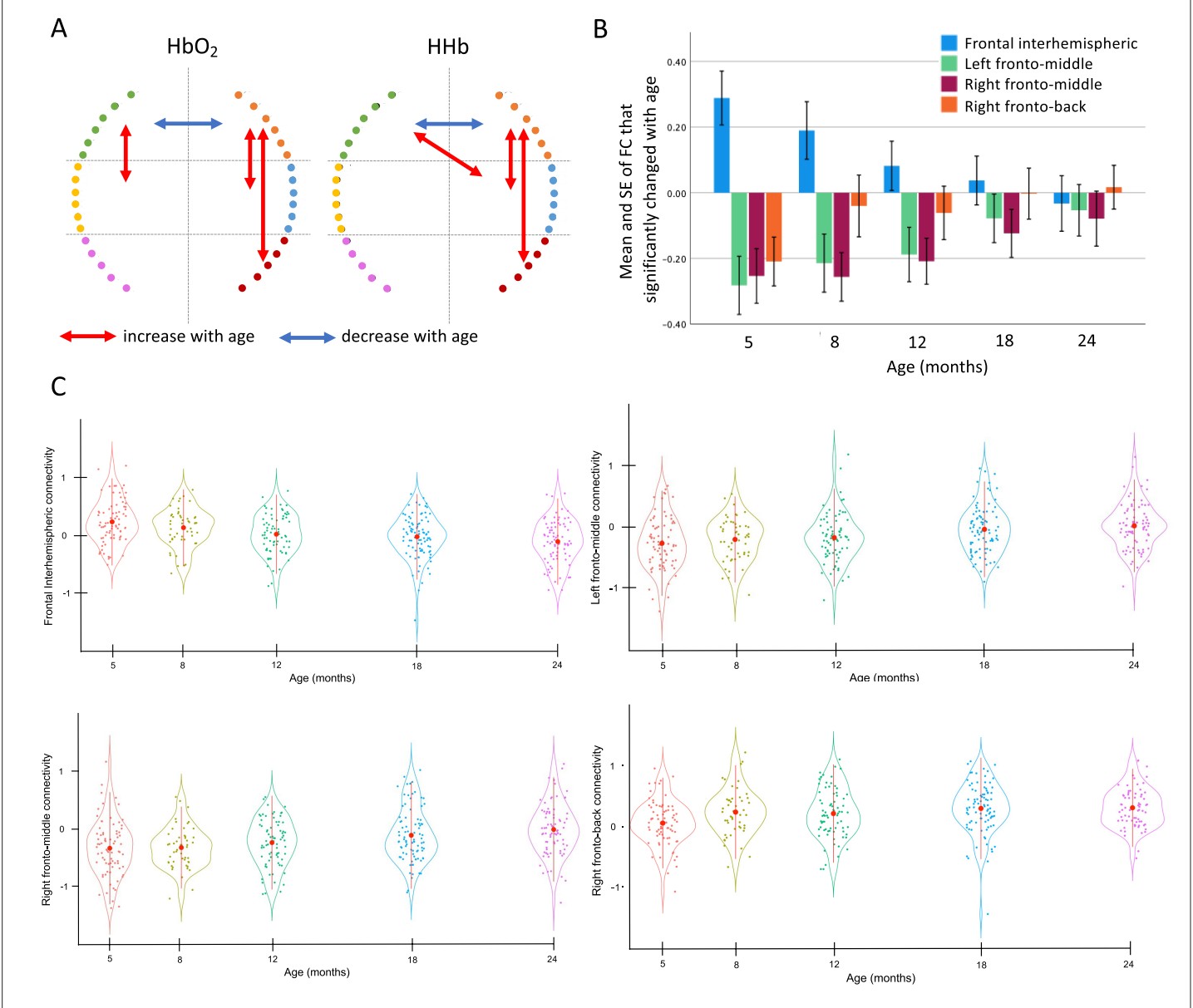

**Figure 2.** Linear mixed models results showing FC that displayed a statistically significant change with age. (**A**) Significant results of the linear mixed model, blue indicates connections that decreased with age, red indicates connections that increased with age. (**B**) Mean and standard error of the mean (SE) of the functional connections that changed with age ($HbO_2$). Error bars are 1 SE. (**C**) Violin plot showing the mean ± SD (red circles and lines) and the individual variability (coloured dots) of the FC that showed a change with time ($HbO_2$).

## Early functional connectivity predicts cognitive flexibility at preschool age

To investigate whether FC early in life predicted cognitive flexibility at preschool age, we used multiple regression of FC across the first 2 years of life against later cognitive flexibility in preschoolers at 3 and 5 years. As per the analysis above, we focused on only those FC that showed a statistically significant change with age. Our results showed that frontal interhemispheric homotopic FC at 5 months ($F(1,38)=4.21$, $p=0.047$, $R^2=0.102$), left frontal-middle ($F(1,33)=7.86$, $p=0.009$, $R^2=0.197$) and right frontal-posterior FC at 12 months ($F(1,38)=4.82$, $p=0.034$, $R^2=0.115$) positively predicted cognitive flexibility in young preschoolers. Right frontal-posterior FC at 8 months negatively predicted cognitive flexibility in young preschoolers ($F(1,32)=5.6$, $p=0.024$, $R^2=0.153$). Frontal interhemispheric homotopic FC at 18 months ($F(1,45)=4.82$, $p=0.030$, $R^2=0.115$), and right frontal-posterior FC at 24 months

**Table 1.** FC that significantly changed with age.

Results are displayed in terms of estimated betas, standard errors and p values that survived Bonferroni correction. Regressions that survived Bonferroni correction for multiple comparisons are in bold.

| FC | F | p | Baseline (5 months) Betas (SE), p | 5–8 change Betas (SE), p | 5–12 change Betas (SE), p | 5–18 change Betas (SE), p | 5–24 change Betas (SE), p |
|---|---|---|---|---|---|---|---|
| HbO₂ | | | | | | | |
| Frontal interhemispheric | 11.03 | <0.001 | 0.28 (0.03),<0.001 | –0.09 (0.06), 0.12 | **–0.20 (0.05),<0.001** | **–0.25 (0.05),<0.001** | **–0.33 (0.05),<0.001** |
| Left fronto-middle | 5.8 | <0.001 | –0.28 (0.04),<0.001 | 0.06 (0.06), 0.293 | 0.09 (0.05), 0.102 | **0.19 (0.05), <0.001** | **0.23 (0.05), <0.001** |
| Right fronto-middle | 4.86 | <0.001 | –0.70 (0.29), 0.018 | 0.01 (0.05), 0.982 | 0.05 (0.05), 0.288 | **0.13 (0.05), 0.007** | **0.19 (0.05), <0.001** |
| Right frontal-posterior | 5.52 | <0.001 | –0.20 (0.03),<0.001 | **0.16 (0.06), 0.005** | **0.14 (0.05), 0.006** | **0.20 (0.05), <0.001** | **0.22 (0.05), <0.001** |
| HHb | | | | | | | |
| Frontal interhemispheric | 6.59 | <0.001 | 0.32 (0.03),<0.001 | –0.09 (0.06), 0.107 | **–0.22 (0.05),<0.001** | **–0.23 (0.05),<0.001** | **–0.22 (0.05),<0.001** |
| Cross left fronto-right middle | 5.17 | <0.001 | –0.32 (0.04),<0.001 | 0.06 (0.06), 0.328 | 0.07 (0.05), 0.188 | **0.19 (0.05), <0.001** | **0.22 (0.05), <0.001** |
| Right fronto-middle | 5.55 | <0.001 | –0.78 (0.32), 0.036 | 0.01 (0.06), 0.86 | 0.05 (0.05), 0.301 | **0.16 (0.05), 0.002** | **0.21 (0.05), <0.001** |
| Right frontal-posterior | 5.38 | <0.001 | –0.24 (0.04),<0.001 | 0.10 (0.06) 0.073 | 0.08 (0.05) 0.133 | **0.23 (0.05), <0.001** | **0.02 (0.05), <0.001** |

($F(1,49)$=4.85, p=0.032, $R^2$=0.092) positively predicted cognitive flexibility in older preschoolers. Left front-middle FC at 18 months negatively predicted cognitive flexibility in older preschoolers ($F(1,45)$=5.72, p=0.021, $R^2$=0.115). While these associations between early functional connectivity and cognitive flexibility at preschool age show promise, none of these survived FDR correction for multiple comparisons (*Figure 3B* and *Appendix 1—table 2*), and so should be considered as preliminary and useful for hypothesis generation for future studies. We also explored whether changes in growth and changes in functional connectivity between 5 and 24 months were associated with cognitive flexibility at preschool age, but we did not find any significant association (*Appendix 1—table 3* and *Appendix 1—table 4*).

## Discussion

This study investigated how early adversity via undernutrition drives brain functional connectivity throughout the first 2 years of life and how these early functional connections are associated with cognitive flexibility at preschool age. To capture the rapid neural development that takes place during early years in community LMIC settings, we used fNIRS to assess brain connectivity (*Gao et al., 2009*; *Gao et al., 2015*; *Fransson et al., 2007*; *Bulgarelli et al., 2020b*). Additionally, we recorded brain activity while infants were awake, thus making the results more comparable with those from adult studies. Our results show: (i) healthy growth trajectories during early infancy are crucial for healthy FC at 24 months; (ii) our cohort of Gambian infants exhibit atypical developmental trajectories of functional connectivity (FC) in over the first 2 years of life; and (iii) FC during the first 2 years of life may predict cognitive flexibility at preschool age.

First, we found that while left and right intrahemispheric fronto-middle and right frontal-posterior FC increased with age, frontal inter-hemispheric FC decreased with age. Additionally, we observed long-range right frontal-posterior FC increased with age and exhibited positive correlations at 24 months, as consistent with previous infant studies (*Damaraju et al., 2014*; *Bulgarelli et al., 2020b*). Previous findings from typically developing infants measured in the US showed that long-range FC gradually strengthened with age, indicating maturing of functional networks that span distant brain

**Table 2.** Results of the regression analyses of the effect of ΔWLZ on FC at 24 months.

Significant positive associations are in green, significant negative associations are in orange, and non-significant (NS) associations are in blue; * indicates regressions that are still significant after correcting for HCAZ at 7/14 days and ** indicates regressions that are still significant after correcting for neonatal HCAZ and WLZ. Regressions that survived FDR correction for multiple comparisons are in bold.

**Frontal interhemispheric FC at 24 months**

| ΔWLZ | birth | 1 month | 5 months | 8 months | 12 months | 18 months | 24 months |
|---|---|---|---|---|---|---|---|
| birth | | NS | $F_{(1,68)}=8.69$, $P=0.004^*$, $R^2=0.102$ | $F_{(1,67)}=7.65$, $P=0.007^*$, $R^2=0.104$ | $F_{(1,69)}=10.4$, $P=0.002^{**}$, $R^2=0.121$ | $F_{(1,68)}=10.98$, $P=0.001^{**}$, $R^2=0.142$ | $F_{(1,59)}=5.40$, $P=0.024^*$, $R^2=0.085$ |
| 1 month | | | $F_{(1,77)}=5.59$, $P=0.021^{**}$, $R^2=0.069$ | NS | $F_{(1,78)}=4.74$, $P=0.032^{**}$, $R^2=0.058$ | $F_{(1,75)}=4.45$, $P=0.038^{**}$, $R^2=0.057$ | NS |
| 5 months | | | | NS | NS | NS | NS |
| 8 months | | | | | NS | NS | NS |
| 12 months | | | | | | NS | NS |
| 18 months | | | | | | | NS |
| 24 months | | | | | | | |

**Left fronto-middle FC at 24 months**

| ΔWLZ | birth | 1 month | 5 months | 8 months | 12 months | 18 months | 24 months |
|---|---|---|---|---|---|---|---|
| birth | | NS | NS | $F_{(1,67)}=4.07$, $P=0.048^{**}$, $R^2=0.058$ | NS | NS | $F_{(1,59)}=4.24$, $P=0.044^{**}$, $R^2=0.068$ |
| 1 month | | | NS | NS | NS | NS | NS |
| 5 months | | | | NS | NS | NS | NS |
| 8 months | | | | | NS | NS | NS |
| 12 months | | | | | | NS | NS |
| 18 months | | | | | | | NS |
| 24 months | | | | | | | |

**Right fronto-middle FC at 24 months**

| ΔWLZ | birth | 1 month | 5 months | 8 months | 12 months | 18 months | 24 months |
|---|---|---|---|---|---|---|---|
| birth | | NS | NS | $F_{(1,67)}=4.74$, $P=0.033$, $R^2=0.067$ | NS | $F_{(1,68)}=5.29$, $P=0.024^{**}$, $R^2=0.073$ | NS |
| 1 month | | | $F_{(1,77)}=4.26$, $P=0.042^{**}$, $R^2=0.053$ | $F_{(1,75)}=6.66$, $P=0.012^{**}$, $R^2=0.083$ | NS | $F_{(1,75)}=5.33$, $P=0.024^{**}$, $R^2=0.067$ | NS |
| 5 months | | | | NS | NS | NS | NS |
| 8 months | | | | | NS | NS | NS |
| 12 months | | | | | | NS | NS |
| 18 months | | | | | | | NS |
| 24 months | | | | | | | |

**Right frontal-posterior FC at 24 months**

| ΔWLZ | birth | 1 month | 5 months | 8 months | 12 months | 18 months | 24 months |
|---|---|---|---|---|---|---|---|
| birth | | $F_{(1,66)}=8.26$, $P=0.006^{**}$, $R^2=0.113$ | NS | NS | NS | NS | NS |
| 1 month | | | $F_{(1,69)}=4.86$, $P=0.031^*$, $R^2=0.067$ | $F_{(1,68)}=6.98$, $P=0.01^{**}$, $R^2=0.094$ | $F_{(1,71)}=8.86$, $P=0.004^{**}$, $R^2=0.112$ | $F_{(1,67)}=7.72$, $P=0.007^{**}$, $R^2=0.105$ | $F_{(1,57)}=7.03$, $P=0.01^{**}$, $R^2=0.112$ |

*Table 2 continued on next page*

*Table 2 continued*

| Right frontal-posterior FC at 24 months | | | | | |
|---|---|---|---|---|---|
| 5 months | | NS | NS | NS | NS |
| 8 months | | | NS | NS | NS |
| 12 months | | | | NS | NS |
| 18 months | | | | | NS |
| 24 months | | | | | |

regions (*Smyser et al., 2011*; *Damaraju et al., 2014*). Moreover, the presence of inter-hemispheric connections are known to indicate healthy development of FC (*Ferradal et al., 2016*) consistent with the continuous development of the corpus callosum from infancy until early adulthood (*Luders et al., 2010*). Conversely, decreased inter-hemispheric FC has been observed where development can follow a more divergent and/or heterogeneous path, such as in autism (*Anderson et al., 2011*). The observed decrease in frontal inter-hemispheric FC with increasing age may be due to the exposure to early life undernutrition adversity. We acknowledge that differences in FC could also be attributed to other environmental and cultural disparities between high-resource and low-resource settings, and future studies are needed to explore this further. Finally, our results indicated that the Gambian infant bilateral fronto-middle FC exhibited anticorrelation at five months, and then gradually became less negative with age. This pattern suggests that this fronto-middle FC may become in sync later after the second year of life. The fNIRS array covers regions over the front and middle cortical regions, putatively belonging to the default mode network and fronto-parietal network, which are normally

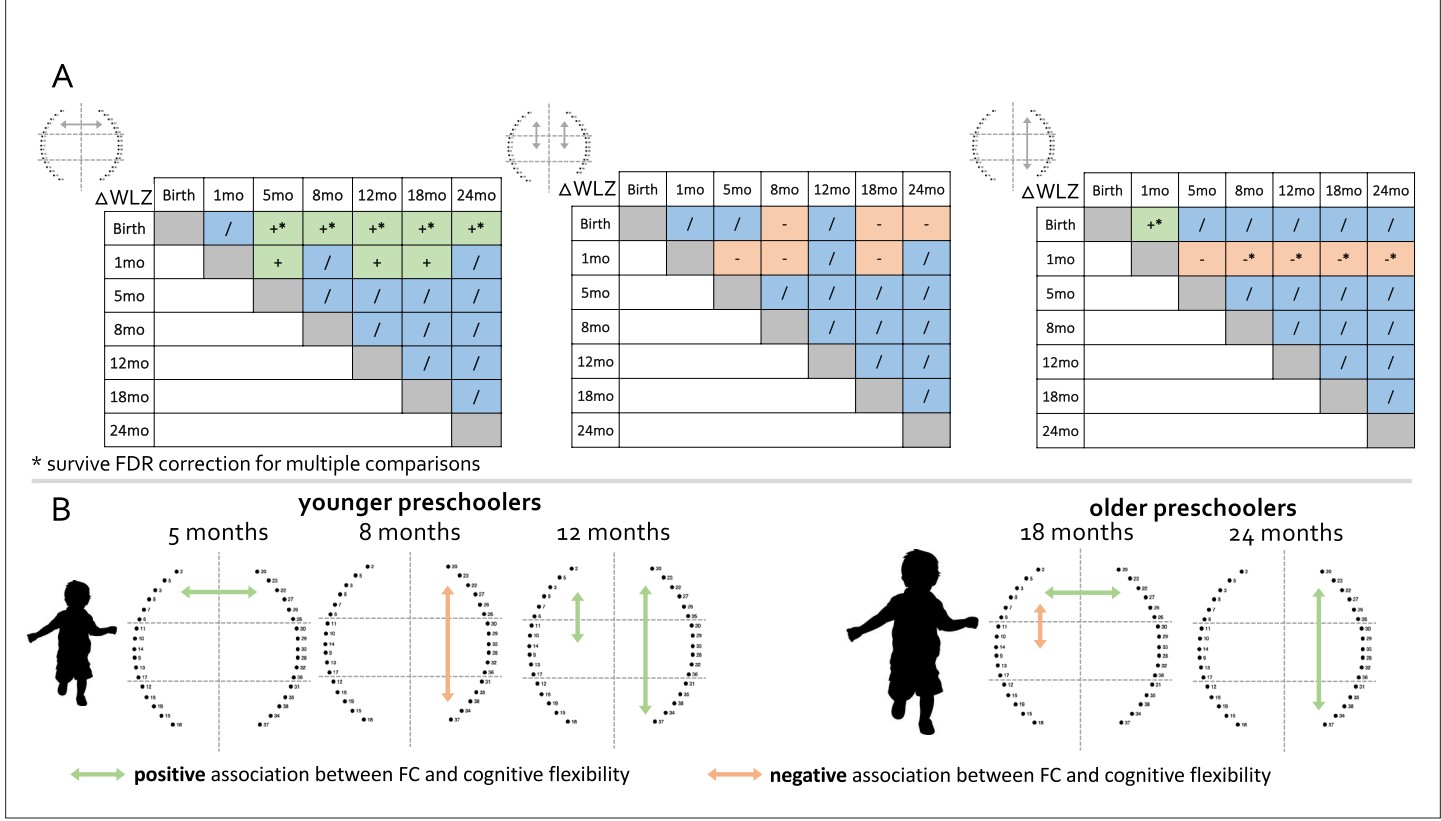

**Figure 3.** Associations between FC, early growth and later cognitive flexibility. (**A**) Significant positive associations are in green, significant negative associations are in orange, and non-significant associations are in blue. * indicates regressions that survived FDR correction for multiple comparisons. (**B**) Schematic representations of the early FC connections shown to predict cognitive flexibility in preschoolers. Significant positive associations are in green, significant negative associations are in orange (these results did not survive corrections for multiple comparisons).

anticorrelated (*Fox et al., 2005*), suggesting that the earlier observed anticorrelation pattern might indeed be adaptive. Importantly, these results remained significant even without GSR, indicating that our findings are not solely driven by preprocessing choices. While the use of GSR in FC studies remains debated (*Murphy and Fox, 2017*), in the absence of short channels (which are difficult to use reliably with infants *Emberson et al., 2016*) and external physiological measures, applying GSR represented the most appropriate preprocessing option. In fact, failure to correct for systemic physiological fluctuations can, in fact, lead to artificially elevated connectivity estimates in fNIRS data (*Abdalmalak et al., 2022*). While further investigations are needed to confirm these findings, these results highlight a general disruption of the integration/segregation process of network development among the infants in our study (*Fair et al., 2007*).

Second, a central goal of the current work was to investigate whether infant nutritional status, as measured by change in weight for length z-scores (ΔWLZ), was associated with functional connectivity at 24 months. First, we observed that ΔWLZ between neonatal time points and older ages positively predicted frontal interhemispheric homotopic FC at 24 months, which supports the idea that the decreasing interhemispheric connectivity observed with age might be atypical, as greater increases in ΔWLZ were found to be positively associated with stronger interhemispheric FC. This is consistent with research showing an impact of nutritional status on long-range connections and brain networks development (*Talukdar et al., 2019*; *Algarin et al., 2017*; *Zamroziewicz et al., 2017*) and the decreased long-range FC found in preterm babies with faltering growth (*Grieve et al., 2008*; *Padilla et al., 2017*; *Duerden et al., 2021*). A common feature of the observed associations between ΔWLZ and FC is that they were statistically significant only when ΔWLZ was calculated including measures collected neonatally (i.e., at birth or one month of age). Changes in growth later in development (i.e. ΔWLZ between 5, 8, 12, or 18 months and older ages) did not show statistically significant impacts on any of the FC assessed. This provides support for the hypothesis that undernutrition during the first months of life is more impactful on brain development than undernutrition occurring later in infancy and early childhood. While the impact of undernutrition on brain development has been documented in LMICs (*Xie et al., 2019*), herein, we provided empirical evidence that growth faltering specifically in infants younger than 5 months of age impacts observable development of functional brain networks in the second year of life. Future studies may be needed to pinpoint which specific brain networks are impacted. This result also suggests that early undernutrition has a lasting impact on subsequent cognitive skills, even in children who show subsequent catch-up growth.

The other primary aim of this study was to investigate whether early FC development was related to cognitive flexibility in preschoolers at 3 and 5 years of age. Long-range FC (interhemispheric homotopic and frontal-posterior) positively predicted performance in the cognitive flexibility task both in younger and older preschoolers. This is consistent with the fact that the integration of distant brain regions fosters cognitive development, especially in the prefrontal cortex (*Case, 1992*; *Hearne et al., 2016*), possibly driven by maturation within the fronto-parietal network that has been widely documented as a primary neural underpinning of cognitive flexibility (*Uddin, 2021*). Additionally, left fronto-middle FC at 12 months positively predicted cognitive flexibility in young preschoolers, but the same connection at 18 months negatively predicted cognitive flexibility in older preschoolers. This shift in direction of the relationship between FC and cognitive flexibility might indicate the shift between short-range and long-range FC during development, where short-range FC still promotes cognitive development early in life, while becoming less impactful as the child ages possibly because children develop a wider range of strategies to support cognitive flexibility as they age. A recent review on the neural underpinnings of cognitive flexibility highlighted that right-lateralised activations underlying this skill increased with age, while activations over the left hemisphere decreased with age (*Dajani and Uddin, 2015*). Our findings are consistent with this view, suggesting an increase in functional specialisation and segregation with development. While our results are consistent with previous studies, we acknowledge that the significant associations between early FC and later cognitive flexibility do not withstand multiple comparisons. Therefore, we encourage future studies that may replicate these findings with a larger sample.

Some limitations of the current study merit discussion. Although the BRIGHT sample is one of the largest longitudinal neuroimaging studies in LMIC to assess longitudinally this many age points during the first 2 years of life, the sample size may still be underpowered to assess the modest effects of the regression models. Second, additional factors may affect FC that were not considered in our

models. Applying more advanced statistical modelling methods and structural equation modelling analyses may provide greater insight with further investigations in contexts of adversity and, in turn, establish which outcomes are predicted by FC. That said, results from the regression models were statistically robust, as most survived FDR correction for multiple comparisons, and were significant after including HCZ at 7–14 days and WLZ at birth in the models. It is also worth noting that in line with previous FC fNIRS studies (*Bulgarelli et al., 2020b*; *White et al., 2009*), linear mixed model results on the HbO$_2$ and the HHb signals showed consistency in the FC changes with age measured in the two chromophores, and similar trajectories when removing the global signal regression step from the pre-processing, suggesting that the results were reliable and statistically robust. Future collaborations between projects studying the impact of adversity on brain development in both high resource and LMIC settings are crucial to advance the field and therefore inform efficient strategies – including the timing - of intervention. Even given the large sample in our study, we were underpowered to test for group comparisons between sets of infants with distinct undernutrition growth profiles, for example infants with early poor growth that later resolved and infants with standard growth early that had a poor growth later. We were also underpowered to test the associations between early growth and FC on clinically undernourished infants (defined as having ΔWLZ two standard deviations below the mean). However, our findings provide a solid foundation for future research on the impacts of undernutrition during the first months of life on later brain health and cognitive abilities throughout childhood.

In conclusion, herein we showed that, as expected, positive growth trajectories during the first few months of life were positively associated with stronger interhemispheric frontal FC and negatively associated with bilateral short-range FC. Interestingly, while Gambian infants had expected increasing left and right intrahemispheric fronto-middle and right frontal-posterior FC with age, frontal inter-hemispheric FC decreased with age, which is inconsistent with extant data from other populations studied in Europe and the US. Of particular note, growth during the first few months after birth showed a significant impact on brain health, as measured with FC, while later growth changes were not related to variability in FC. Moreover, long-range FC predicted the performance on a cognitive flexibility task at preschool age. Taken together, our results show the impact that early growth has on functional brain development, which in turn affects cognitive flexibility in preschoolers. Finally, this study emphasizes the importance of the timing of interventions, for which benefits are visible at preschool age and beyond.

## Materials and methods

### Experimental design

Our study investigated the developmental trajectory of brain health as assessed with functional connectivity (FC) in Gambian infants and its relationship with early growth, as a proxy for nutritional status, and later cognitive outcomes. To do so, we estimated FC from data acquired with fNIRS longitudinally at five time points between age five and 24 months in the context of the BRIGHT Project. The BRIGHT project sought to provide brain function-for-age curves from the UK and Gambian infants and to gain insight into the effects that issues related to living in low-resource settings may have on brain development. We then tested for relationships between brain FC at age 24 months with measures of early growth, as indexed by changes in weight-for-length z-scores (reflecting body weight in proportion to attained growth in length) at 1 month of age, and at each of the four subsequent visits (details provided below). We also tested for associations between brain FC at each time point with later cognitive outcome, as indexed by cognitive flexibility assessed in 3- and 5-year-old preschool-age children (*Figure 1A*). Ethical approval was granted by the joint Gambia Government - MRC Unit the Gambia Ethics Committee, and written informed consent was obtained from all parents prior to participation.

### Participants

We used data from N=204 infants recruited in rural community settings of The Gambia as part of the BRIGHT project. Ethical approval was given by the joint Gambia Government - MRC Ethics Committee (SCC 1351) and the Scientific Coordinating Committee at the MRC Unit The Gambia. Additional approval was granted for the BRIGHT Kids follow up (Project reference 22737). Informed consent was obtained in writing, or via thumbprint if individuals were unable to write, from all parents/carers prior

**Table 3.** Demographic information per each age.

| Age | Age in days (mean ± SD) | Sex (M, F) | WLZ (mean ± SD) | HCZ (mean ± SD) |
| --- | --- | --- | --- | --- |
| 5 months | 158.92±9.91 | 45,42 | –0.24±0.95 | –0.72±0.92 |
| 8 months | 249.34±17.97 | 28,25 | –0.24±1.01 | –0.81±0.90 |
| 12 months | 372.92±14.75 | 42,40 | –0.58±1.05 | –0.95±0.95 |
| 18 months | 560.37±24.38 | 48,49 | –0.85±0.91 | –0.82±0.93 |
| 24 months | 746.61±23.25 | 49,47 | –0.44±0.99 | –0.80±0.97 |

to participation. The study was a prospective, longitudinal study with fNIRS assessments planned for awake infants at ages 5, 8, 12, 18, and 24 months. All infants were born full term (37–42 weeks gestation). fNIRS data were additionally acquired from 1-month-old infants while asleep. However, as sleep stages may impact connectivity measures (*Mitra et al., 2017*), these data are not part of the current analysis. Reasons for exclusion from the fNIRS analysis were: (i) fussiness; (ii) experimental errors (missing pictures of the headgear placement or any other technical issue); (iii) poor headgear placement (see supplementary information for *Assessment of headgear placement*); (iv) low-quality data (more than 40% of the channels excluded as previously done *Bulgarelli et al., 2020a*; *Collins-Jones et al., 2021*; *Katus et al., 2023*; *Lloyd-Fox et al., 2019*). *Appendix 1—table 5* summarises the details of the included and excluded participants for each visit, showing an attrition rate within the standard range for infant fNIRS studies (*Baek et al., 2023*; see *Table 3* for the demographic information of the participants included in the analyses at each age).

## Anthropometric measures

Measurements of length, weight, and head circumference (HC) were collected on all infants at birth, at 7/14 days, 1 month, and in all visits from 5 to 24 months, by trained field workers using calibrated tools. Length was measured using a Harpenden Infantometer length board (Holtain Ltd) to a precision of 0.1 cm. Weight was obtained using an electronic baby scale (Model 336, SECA) to a precision of 0.01 kg. Finally, HC was measured around the maximum circumference of the head (forehead to occiput) using stretch-proof measuring tape (Model 201, SECA) to the nearest 0.1 cm. Each measure was taken in triplicate and the mean of the three measures was used in analyses.

## fNIRS data acquisition

fNIRS data were collected using the NTS topography system (Gowerlabs Ltd., London, UK). This system uses two continuous wavelengths of near-infrared light (780 and 850 nm) to detect changes in $HbO_2$ and HHb concentrations, using a sampling rate of 10 Hz (*Everdell et al., 2005*).

Infants were assessed using a custom-built fNIRS headgear consisting of two arrays, including 14 sources and 12 detectors to create a total of 34 channels, covering bilateral frontal, inferior-frontal and temporal regions, with a source-detector separation of 2 cm (*Figure 1B*). Infants underwent several fNIRS tasks as part of the BRIGHT project, focusing on social processing, deferred imitation, and habituation response. Therefore, brain regions for data acquisition were chosen to maximize the likelihood of recording meaningful data for all tasks (*Lloyd-Fox et al., 2024*).

For each infant, head measurements were taken to enable the alignment of the headgear with the 10–20 placement coordinates (*Lloyd-Fox et al., 2014*). Over the front of the head, the headgear was placed so that a vertical red line marking the centre of the band aligned with the nasion with the silicon band laying just above, and in line with, the eyebrows. The headgear was placed so that the reference optode (the third lower optode from the back) was placed over the tragus.

To model for the headgear placement offline, photographs of the participants (one from the front and one from either side) were taken after the headgear was placed and again at the end of the fNIRS session. To assess the headgear placement, photographs of the participants had to be sufficiently good in quality (i.e. not blurred) and with the camera pointing at the ear, at a position directly perpendicular to the infant's anatomical reference on the side of the head being photographed.

Infants were excluded from further analysis if the band was excessively high over the front above the eyebrows, by checking the displacement of the lower edge of the band with respect to the

eyebrows with a 'traffic-light' code (red, orange, green). Three experimenters independently classified the infants' frontal headgear placement, and discrepancies were discussed until agreement was reached. Channels of those infants with valid placement over the front but horizontal lateral displacement between the ideal and the actual position of the reference (optode marked in green in *Figure 1B*) equal to or greater than 1.6 cm were renumbered, so that each channel was shifted either backward or forward one full channel location in space.

At each visit, the fNIRS data acquisition took place while infants sat on their parent's lap. The parent was instructed to refrain from interacting with the infant during the stimuli presentation. Videos were of male and female Gambian adults singing nursery rhymes in Mandinka (for 70 s) and videos of toys in action (for 60 s), video blocks alternated, with each type shown to the infant twice, with the aim of keeping the infant awake, calm, attentive and as still as possible. The functional connectivity (FC) paradigm formed part of a greater battery of fNIRS paradigms that ran in a continuous and consistent order across all infants. The fNIRS FC acquisition lasted for a maximum of 520 s (two rounds of 260 s each) or was stopped as soon as the participant became fussy. Stimuli were presented using a MATLAB custom-written stimulus presentation framework, Task Engine (sites.google.com/site/taskenginedoc) and Psychtoolbox on an Apple Macintosh computer.

## fNIRS preprocessing and data-analysis

Data analyses were carried out using in-house programmes developed in MATLAB (MathWorks, Natick, MA; *Appendix 1—figure 4* summarises fNIRS data preprocessing steps). Stringent multi-level data-quality assessments were used to ensure all data included in the group were of adequate quality. First, low-quality channels based on physiological indicators of quality were pruned using QT-NIRS (https://github.com/lpollonini/qt-nirs, *Pollonini, 2026*). The software takes advantage of the fact that the cardiac pulsation is also recorded in addition to haemodynamic activity associated with brain-driven variance including task-based responses and those contributing to functional connectivity (*Tachtsidis and Scholkmann, 2015*), and quantifies its strength in the spectral and temporal domains with two measures: the Scalp Coupling Index (SCI) and the Peak Spectral Power (PSP; for more details, see *Hernandez and Pollonini, 2020*). For each infant, data quality was assessed channel-by-channel, and SCI and PSP were calculated with sliding 3 s windows (threshold SCI = 0.70, threshold PSP = 0.1, empirically defined and based on *Hernandez et al., 2022*; *Hernandez and Pollonini, 2020*). Channels that had both SCI and PSP below threshold for more than 70% of the windows (*Hernandez and Pollonini, 2020*) were excluded from further pre-processing steps.

Raw intensity data were converted to optical density via a logarithm after dividing by the temporal mean of each channel (*hmrIntensity2OD.m* function from Homer2 tool *Huppert et al., 2009*) and band-pass filtered (0.009–3 Hz; *hmrBandpassFilt.m* function from Homer2 *Huppert et al., 2009*). Using first a wide filter leaves in the signal contamination from the high bandwidth information (i.e. respiration, pulse, etc.) that we aim to regress out next. To help manage the systemic arterial signal that is well known to contaminate fNIRS signals (*Gregg et al., 2010*), we regressed out the mean of the signals across the array, a method also known as global signal regression (GSR; *Yücel et al., 2021*; *Murphy and Fox, 2017*) (Results of analysis performed also without GSR are reported in the Supplementary Materials). Hereafter, optical density data underwent a second bandpass filter (0.009–0.08 Hz), as previously described (*Eggebrecht et al., 2014*). Motion artefacts were then rejected using global variance of temporal derivatives (GVTD, STD = 5) in the NeuroDOT toolbox (https://www.nitrc.org/projects/neurodot/; *Sherafati et al., 2020*; *Appendix 1—figure 5* shows how we tested the effect of different STD thresholds of GVTD on data inclusion). To further manage potential motion contamination of the data, we additionally removed five seconds before and after each motion artefact, and only chunks of data that were at least 20 s long were retained.

Optical density data were converted to relative concentrations of haemoglobin using the modified Beer–Lambert law (*hmrOD2Conc.m* function from Homer2 *Huppert et al., 2009*). Differential path length factors were calculated (*Scholkmann and Wolf, 2013*) and adapted to wavelengths (780 and 850 nm) and ages (5 months=5.24 and 4.25; 8 months=5.26 and 4.26; 12 months=5.27 and 4.28;18 months=5.30 and 4.30; 24 months=5.32 and 4.32).

To increase the statistical power of our analysis and to reduce the number of multiple comparisons for all statistical analysis to investigate developmental changes in connectivity, we averaged the concentration changes of the channels that survived pre-processing into sections. The sections were

established via the 17 channels of each hemisphere which were grouped into front, middle and back (for a total of six regions) based on a previous co-registration of the BRIGHT fNIRS arrays onto age-appropriate templates (*Collins-Jones et al., 2021*; *Appendix 1—figure 1*). Each section represented a reasonable estimate of an anatomically consistent array coverage across the group, and was used to increase statistical power while minimising errors due to variability in head anatomy and cap placement. For each participant, the Pearson-r correlation matrix between all the sections was calculated for both $HbO_2$ and HHb, resulting in a 6×6 matrix of section-pair correlations. We then applied a Fisher z-transformation on the correlation matrix for further statistical analyses. Infants with at least 250 s of valid data after pre-processing were considered for further analyses. To choose the optimal minimum amount of valid data, per each infant at each time point for both the $HbO_2$ and HHb signal, we correlated the connectivity matrices estimated in the first portion of data with the one in the last portion of data by increasing the seconds of data considered (i.e. the first 60 s with the last 60 s, the first 100 s with the last 100 s, the first 120 s with the last 120 s, etc.). We found that correlating the first and the last 250 ss of valid data after pre-processing provided the highest percentage of infants with strong correlation between the first and the last portion of data, suggesting that FC patterns reached a usable stability after 250 s (*Appendix 1—figure 6*).

## Cognitive flexibility at preschool age

Gambian infants from BRIGHT were cross-sectionally assessed at the age of 3 or 5 years for cognitive flexibility using the 'card sorting' task from the tablet-based Early Years Toolbox (*Howard and Melhuish, 2017*; http://www.eytoolbox.com.au). In this task, children were asked to sort cards (i.e. red rabbits or blue boats) either by colour (red or blue) or shape (rabbits or boats), and to flexibly switch back and forth between these rules (*Milosavljevic et al., 2024*).

Due to disruptions to field work as a result of a political crisis in December 2016 - January 2017 that forced us to pause recruitment, data were collected in two ranges of preschool ages, younger preschoolers (age mean ± SD = 47.96±2.77 months, N=77) and older preschoolers (age mean ± SD = 57.58±2.11 months, N=84; *Lloyd-Fox et al., 2024*).

## Statistical analyses

To investigate developmental trajectories of functional connectivity between 5 and 24 months, we ran linear mixed models (*Oberg and Mahoney, 2007*) which is the standard statistical test to analyse repeated measures and longitudinal data with dropouts (*Cnaan et al., 1997*). Statistical analysis was performed with SPSS Version 28.0 statistic software package. Compared to repeated measures ANOVA, linear mixed models account for within-participant dependence and allow for missing data by using only information from the individual at the other visits (*Krueger and Tian, 2004*). All possible interhemispheric homotopic, intrahemispheric within section, fronto-posterior, and crossed connections between the six regions were inserted as dependent variables in a linear mixed model, for a total of 21 linear mixed models. Functional connectivity was modelled as the dependent variable as a function of age with a random participant effect and random errors. Intercept and age were fixed effects, while within-participant dependence was modelled as a random effect (*Meteyard and Davies, 2020*). This same procedure was used in other longitudinal studies that explored developmental changes over time (*Bulgarelli et al., 2020b*; *Pusponegoro et al., 2017*; *Wierenga et al., 2018*). The linear mixed model included the 132 infants who had valid data for at least two visits (55 infants contributed with data from two visits, 51 infants contributed with data from three visits, 22 infants contributed with data from four visits, four infants contributed with data from all of the five visits). The type of covariance between the observations was specified as Autoregression (AR) as two measures close in time of the same participant are likely to be correlated (*Selig and Little, 2012*). To ensure statistical reliability, significant results from the linear mixed models were corrected for multiple comparisons using Bonferroni correction (p=0.00238).

Anthropometric measures were converted to age and sex adjusted z-scores that are based on World Health Organization Child Growth Standards (*WHO Multicentre Growth Reference Study Group, 2006*). Weight-for-Length (WLZ) and Head Circumference (HCZ) z-scores were computed. Delta WLZ (ΔWLZ) were calculated for various periods (0–1 months, 0–5 months, etc.) by subtracting the z-score at the later age from that at the previous age. The use of delta z-scores as outcome measures allowed us to assess the impact of positive or negative deviation from the expected growth

trajectories. To investigate the impact of undernutrition on FC development, we used ΔWLZ as independent variables in regression analyses on $HbO_2$ (as the chromophore with the highest signal-to-noise ratio) FC at 24 months, our final time point of data collection. To maximise power, we considered only those FC that showed a statistically significant change with age. These analyses were adjusted by WLZ at birth and HCZ at 7/14 days, to more confidently assume that the associations between growth and FC were specific to the impact of change in WLZ postnatally and not confounded by the size or maturity of the infant at birth. We used HCZ measured at 7/14 days and not HCZ at birth as HC measures at birth are unreliable following vaginal delivery. Therefore, HCZ at 7/14 days is a more accurate head measure than HCAZ at birth. To ensure statistical reliability, results from the regression analyses on each FC were corrected for multiple comparisons using false discovery rate (FDR; *Benjamini and Hochberg, 1995*) per each connection investigated, that is 21 possible ΔWLZ values per each connection.

To investigate whether FC early in life predicted cognitive flexibility at preschool age, we regressed later cognitive flexibility against FC that showed a significant change across the first 2 years of life.

## Acknowledgements

We thank the parents and infants who took part in this study without whom this work would not have been possible. We also thank the broader team of staff at the MRC Keneba Field Station for supporting us in the collection of this data. This study was supported by: a Bill and Melinda Gates Foundation Grant OPP1127625; core funding MC-A760-5QX00 to the International Nutrition Group by the Medical Research Council UK; the UK Department for International Development (DfID) under the MRC/DfID Concordant agreement. CB is supported by a Leverhulme Trust Early Career Fellowship (ECF-2021-174). BM is supported by an ESRC Secondary Data Analysis Initiative Grant (ES/V016601/1). SEM and SMC are supported by a Wellcome Trust Senior Research Fellowship award to SEM (220225/Z/20/Z). AE is supported by funding from the National Institute of Mental Health (R01MH122751).

## Additional information

### Group author details

**The BRIGHT Study Team**
**Muhammed Ceesay; Kassa Kora; Fabakary Njai; Andrew Prentice; Mariama Saidykhan**

### Competing interests

Chiara Bulgarelli: is a research consultant for Gowerlabs Ltd., the company that produces the NTS optical topography system used in this work. The other authors declare that no competing interests exist.

### Funding

| Funder | Grant reference number | Author |
| --- | --- | --- |
| Leverhulme Trust | ECF-2021-174 | Chiara Bulgarelli |
| Economic and Social Research Council | ES/V016601/1 | Bosiljka Milosavljevic |
| Wellcome Trust | 10.35802/220225 | Samantha McCann Sophie E Moore |
| National Institute of Mental Health | R01MH122751 | Adam T Eggebrecht |
| Gates Foundation | OPP1127625 | Clare E Elwell Sophie E Moore Sarah Lloyd-Fox |
| Medical Research Council | MC-A760-5QX00 | Sophie E Moore |

| Funder | Grant reference number | Author |
|---|---|---|

The funders had no role in study design, data collection and interpretation, or the decision to submit the work for publication. For the purpose of Open Access, the authors have applied a CC BY public copyright license to any Author Accepted Manuscript version arising from this submission.

## Author contributions

Chiara Bulgarelli, Conceptualization, Data curation, Software, Formal analysis, Validation, Investigation, Visualization, Methodology, Writing – original draft, Writing – review and editing; Anna Blasi, Conceptualization, Data curation, Software, Formal analysis, Supervision, Visualization, Methodology, Writing – review and editing; Samantha McCann, Bosiljka Milosavljevic, Ebrima Mbye, Ebou Touray, Tijan Fadera, Lena Acolatse, Data curation, Investigation, Methodology, Project administration, Writing – review and editing; Giulia Ghillia, Investigation, Methodology, Writing – review and editing; Sophie E Moore, Sarah Lloyd-Fox, Clare E Elwell, Conceptualization, Resources, Supervision, Funding acquisition, Project administration, Writing – review and editing; Adam T Eggebrecht, Conceptualization, Software, Formal analysis, Supervision, Visualization, Writing – original draft, Writing – review and editing

## Author ORCIDs

Chiara Bulgarelli https://orcid.org/0000-0002-6153-137X
Sarah Lloyd-Fox https://orcid.org/0000-0001-6742-9889

## Ethics

Human subjects: Ethical approval was granted by the joint Gambia Government - MRC Unit The Gambia Ethics Committee, and written informed consent was obtained from all parents prior to participation.

Reviewer #1 (Public review): https://doi.org/10.7554/eLife.94194.4.sa1
Reviewer #2 (Public review): https://doi.org/10.7554/eLife.94194.4.sa2
Reviewer #3 (Public review): https://doi.org/10.7554/eLife.94194.4.sa3
Author response https://doi.org/10.7554/eLife.94194.4.sa4

# Additional files

## Supplementary files

MDAR checklist

## Data availability

Data supporting this paper will be made available subject to established data sharing agreements. Access to any data collected during or generated by the BRIGHT project is fully audited, and to ensure data security, is overseen by the data management team in the UK and The Gambia. While data sharing is critically important to maximising the benefit of research, we must also consider the need to protect the confidentiality of this sensitive group (particularly the infants within the mother-infant dyads, who as minors do not consent for themselves). Due to the nature of the data being collected (i.e. collected from a specific geographical location, longitudinal dataset of several datapoints) the majority of the data cannot be fully de-identified under the guidance included in the European General Data Protection Regulation (GDPR). The conditions of our ethics approval do not allow public archiving of pseudonymised study data. The data cannot be fully anonymised due to the nature of combined sources of information, such as neuroimaging, sociodemographic, geographic and health measures, making it possible to attribute data to specific individuals, and hence, falling under personal information, the release of which would not be compliant with GDPR guidelines unless additional participant consent forms are completed. For more details see *Lloyd-Fox et al., 2024*.

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

# Appendix 1

## Supplementary results

Linear mixed model results showing the developmental trajectories of functional connectivity in Gambian infants over the first 2 years of life age (fNIRS pre-processing without global signal regression)

To rule out that the global signal regression (GSR) performed as part of our fNIRS preprocessing might have had an impact on our results, we re-processed the data without this step. All the other steps were kept the same. Hereafter, we reperformed the linear mixed models (LMM) on all the possible 21 interhemispheric homotopic, intrahemispheric within section, fronto-posterior, and crossed connections.

Results on the Fisher-z transformed correlation coefficients (z-RHO scores) on the oxygenated haemoglobin (HbO$_2$) showed that left ($F=10.4$, $p<0.001$) and right fronto-middle ($F=4.82$, $p<0.001$) FC increased with age. Results on the Fisher-z transformed correlation coefficients (z-RHO scores) on the deoxygenated haemoglobin (HHb) showed that frontal interhemispheric FC decreased with age ($F=3.5$, $p<0.002$; *Appendix 1—table 1* and *Appendix 1—figure 2*).

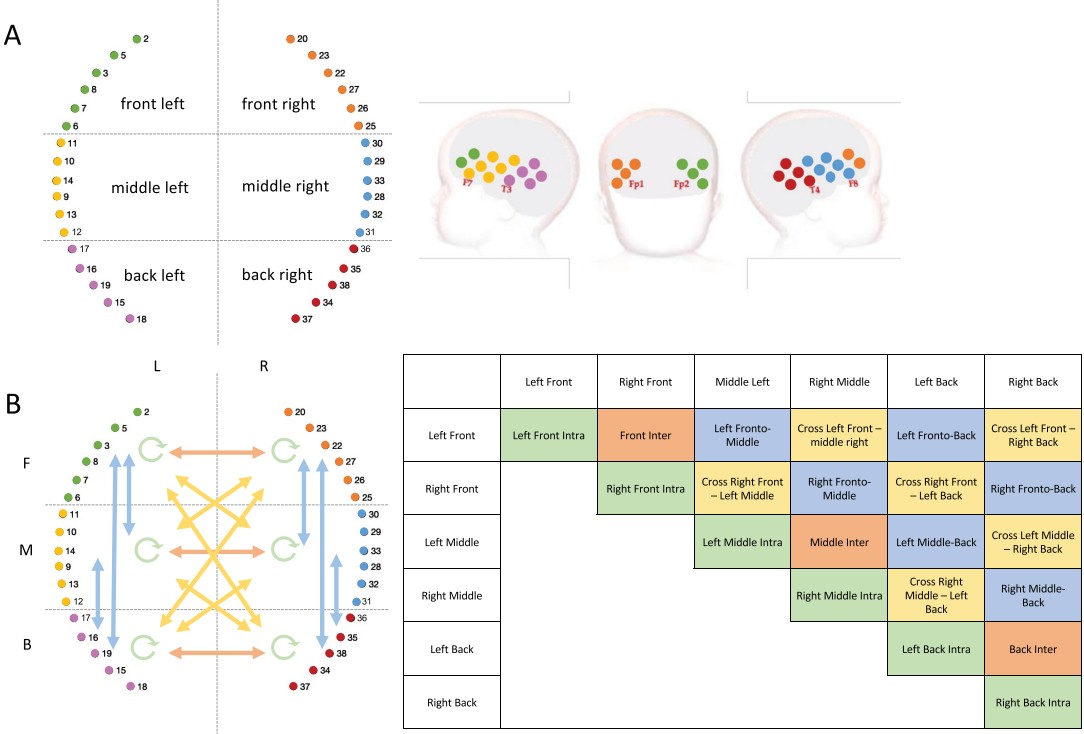

**Appendix 1—figure 1.** Schematic representation of the fNIRS array and the functional connections tested. (**A**) Each dot represents a channel, colours on the left plot correspond to colours on the right on the baby's head. Six sections. (**B**) The 21 connections tested in the linear mixed models. Interhemispheric homotopic connections are in orange (connecting the same regions between hemispheres, that is front left with front right), intrahemispheric connections within section are in green (correlations of channels belonging to the same region), fronto-posterior are in blue (connecting front and middle, middle and back, and front and back regions of the same hemisphere), and crossing interhemispheric connections (interhemispheric non-homotopic, connecting the front and middle, middle and back, and front and back regions of the two hemispheres) are in yellow.

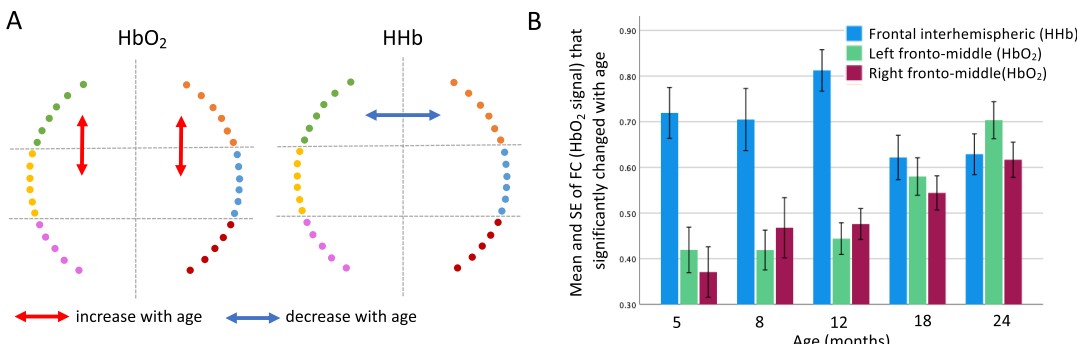

**Appendix 1—figure 2.** Linear mixed models result showing FC that displayed a statistically significant change with age (fNIRS pre-processing without global signal regression). (**A**) Results of the linear mixed model, blue indicates connections that decreased with age, red indicates connections that increase with age. (**B**) Mean and SE of the functional connections that changed with age. Error bars are 1 SE.

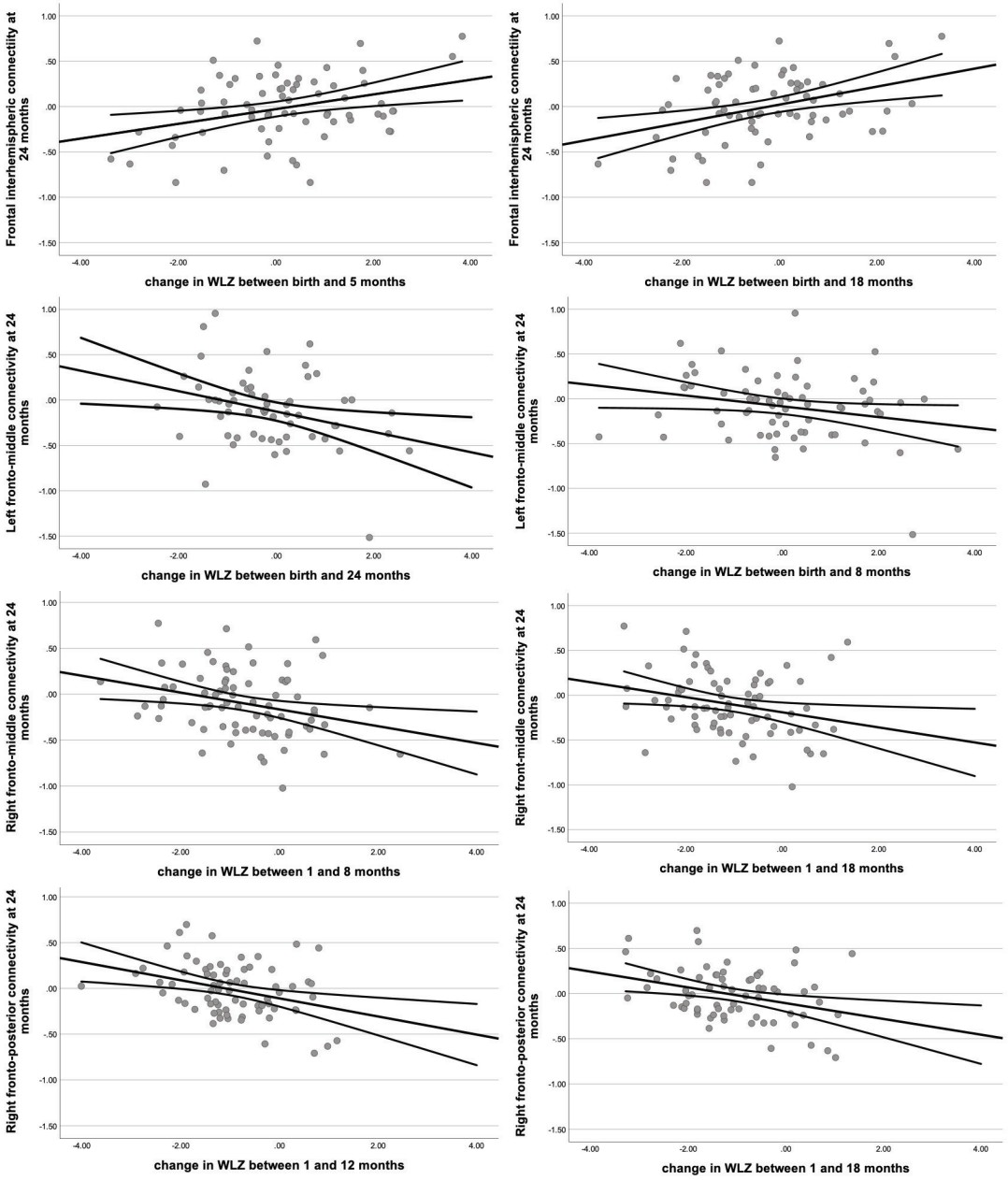

**Appendix 1—figure 3.** Some examples of scatterplots showing the association between ΔWLZ and FC at 24 months. The black lines represent the line of best fit and the 95% confidence intervals.

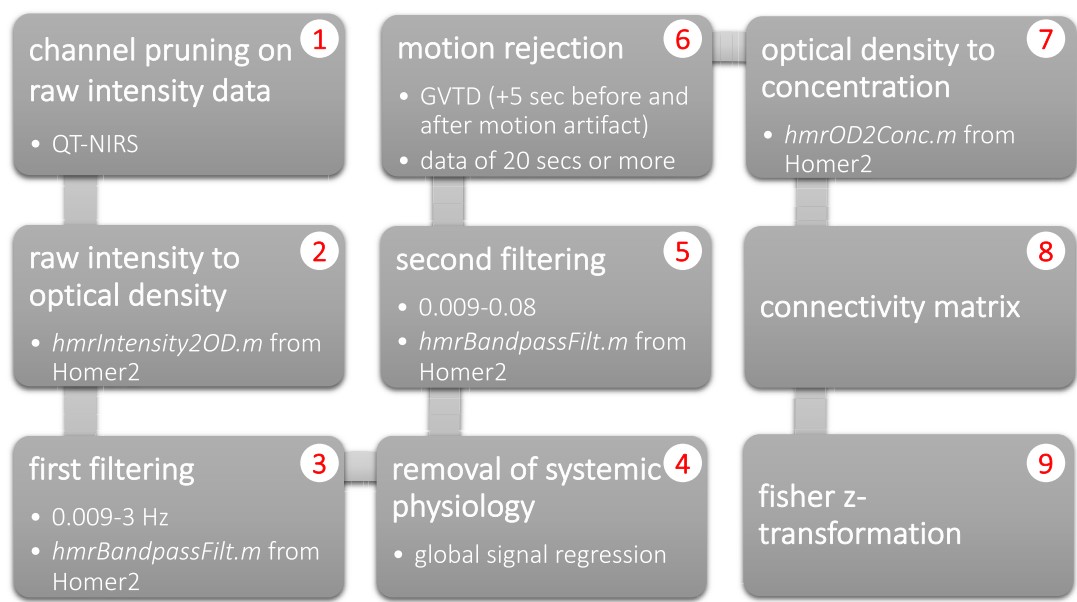

**Appendix 1—figure 4.** fNIRS preprocessing steps. The column 'infants included in the analyses' in *Table 3* refers to those participants whose data survived these preprocessing steps.

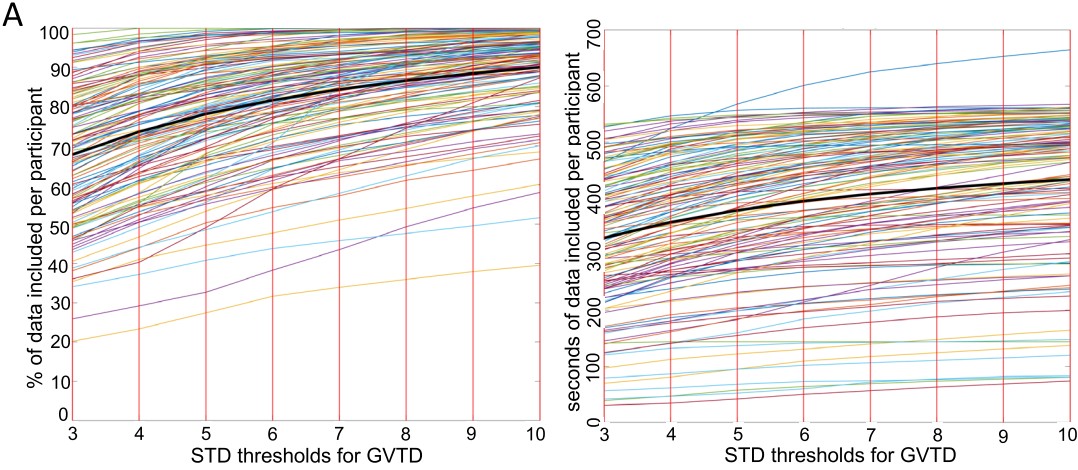

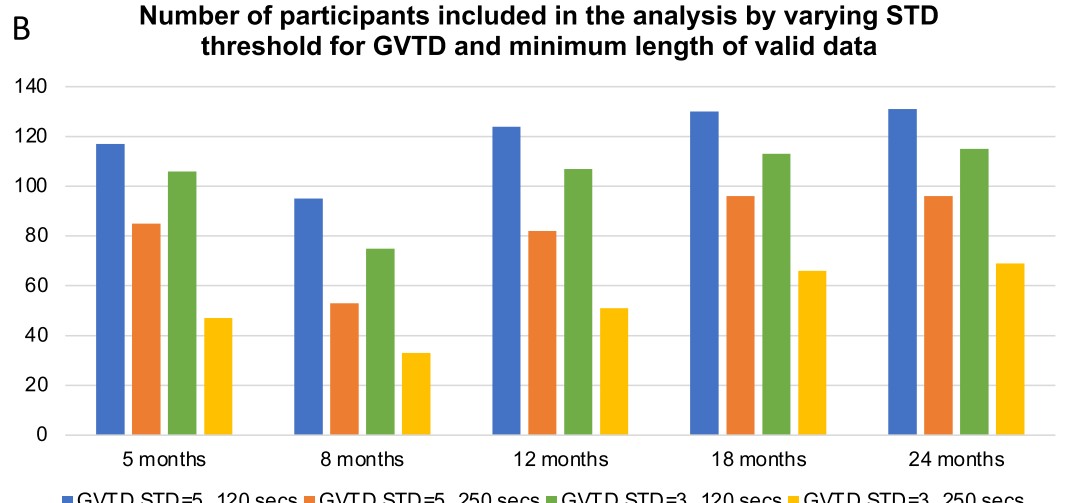

**Appendix 1—figure 5.** The effect of different thresholds for GVTD for motion detection and minimum valid data after pre-processing on data inclusion. (**A**) Percentage of data included (left) and seconds of data included (right) per participant. Each line represents an infant, the black line represents the mean value. These graphs are reported from the 12 months sample as example. (**B**) Number of infants included in the LMM by varying STD threshold for GVTD and minimum length of valid data at different ages.

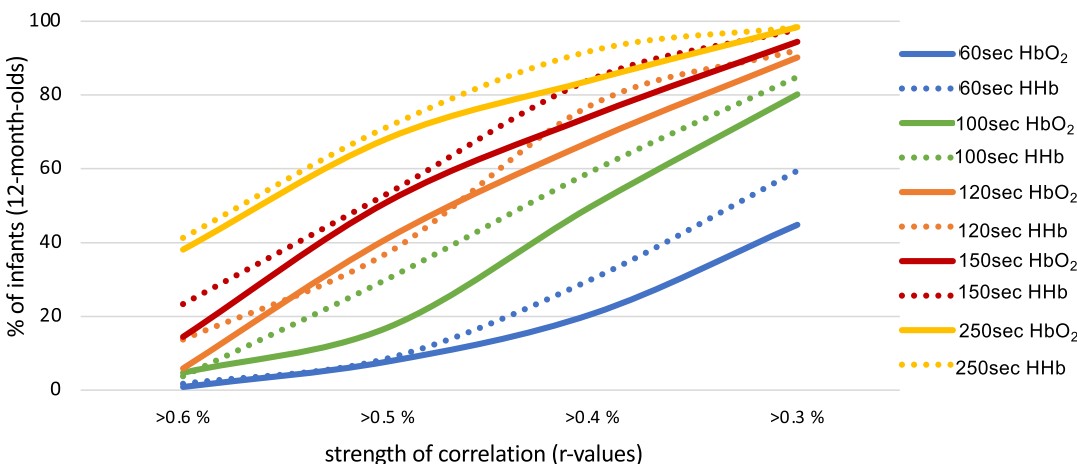

**Appendix 1—figure 6.** Strength of correlation between FC in the first and last portion of data. This graph is reported from the age 12 month sample as example.

**Appendix 1—table 1.** FC that significantly changed with age (fNIRS pre-processing without global signal regression).
Results are displayed in terms of estimated betas, standard errors, and p values.

| FC | $F$ | p | Baseline (5 months) Betas (SE), p | 5–8 change Betas (SE), p | 5–12 change Betas (SE), p | 5–18 change Betas (SE), p | 5–24 change Betas (SE), p |
|---|---|---|---|---|---|---|---|
| HbO$_2$ | | | | | | | |
| Left fronto-middle | 10.4 | <0.001 | −0.55 (0.29),<0.063 | 0.05 (0.05), 0.239 | 0.07 (0.04), 0.125 | 0.16 (0.04),<0.001 | 0.26 (0.04),<0.001 |
| Right fronto-middle | 4.82 | <0.001 | −0.24 (0.28), 0.398 | 0.03 (0.05), 0.535 | 0.05 (0.04), 0.246 | 0.08 (0.04), 0.051 | 0.18 (0.04),<0.001 |
| HHb | | | | | | | |
| Frontal interhemispheric | 3.5 | 0.002 | 0.18 (0.33) 0.591 | −0.07 (0.06), 0.260 | −0.07 (0.05), 0.214 | −0.17 (0.05), 0.002 | −0.17 (0.05), 0.002 |

**Appendix 1—table 2.** Results of the regression analyses of the effect of FC on cognitive flexibility in younger and older preschoolers.
Significant positive associations are in green, significant negative associations are in orange, and non-significant (NS) associations are in blue.

| | | Cognitive flexibility in younger preschoolers | Cognitive flexibility in older preschoolers |
|---|---|---|---|
| | 5 months | F(1,38)=4.21, p=0.047, R2=0.102 | NS |
| | 8 months | NS | NS |
| | 12 months | NS | NS |
| | 18 months | NS | F(1,45)=4.82, p=0.030, R2=0.115 |
| Frontal interhemispheric FC | 24 months | NS | NS |

*Appendix 1—table 2 Continued on next page*

*Appendix 1—table 2 Continued*

| | | Cognitive flexibility in younger preschoolers | Cognitive flexibility in older preschoolers |
|---|---|---|---|
| | 5 months | NS | NS |
| | 8 months | NS | NS |
| | 12 months | F(1,33)=7.86, p=0.009, R2=0.197 | NS |
| | 18 months | NS | F(1,45)=5.72, p=0.021, R2=0.115 |
| Left fronto-middle FC | 24 months | NS | NS |
| | 5 months | NS | NS |
| | 8 months | NS | NS |
| | 12 months | F(1,38)=4.82, p=0.034, R2=0.115 | NS |
| | 18 months | NS | NS |
| Right fronto-middle FC | 24 months | NS | NS |
| | 5 months | NS | NS |
| | 8 months | F(1,32)=5.6, p=0.024, R2=0.153 | NS |
| | 12 months | NS | NS |
| | 18 mo | NS | NS |
| Right frontal-posterior FC | 24 mo | NS | F(1,49)=4.85, *P*=0.032, R2=0.092 |

**Appendix 1—table 3.** Results of the correlational analyses between changes in growth (ΔWLZ) and cognitive flexibility in younger and older preschoolers.

**Cognitive flexibility in younger preschoolers**

| ΔWLZ | birth | 1 month | 5 months | 8 months | 12 months | 18 months | 24 months |
|---|---|---|---|---|---|---|---|
| birth | | r(54)=−0.169, p=0.214 | r(51)=−0.167, p=0.232 | r(51)=−0.249, p=0.072 | r(53)=−0.198, p=0.147 | r(51)=−0.151, p=0.280 | r(41)=−0.241, p=0.102 |
| 1 mo | | | r(64)=0.031, p=0.805 | r(63)=0.001, p=0.993 | r(66)=0.013, p=0.916 | r(62)=0.084, p=0.511 | r(48)=−0.021, p=0.885 |
| 5 mo | | | | r(64)=−0.075, p=0.548 | r(66)=−0.032, p=0.798 | r(63)=0.008, p=0.947 | r(50)=−0.091, p=0.520 |
| 8 mo | | | | | r(66)=−0.044, p=0.719 | r(62)=0.067, p=0.600 | r(50)=−0.041, p=0.775 |
| 12 mo | | | | | | r(64)=0.097, p=0.439 | r(51)=−0.012, p=0.932 |
| 18 mo | | | | | | | r(49)=−0.107, p=0.454 |
| 24 mo | | | | | | | |

**Cognitive flexibility in older preschoolers**

| ΔWLZ | birth | 1 month | 5 months | 8 months | 12 months | 18 months | 24 months |
|---|---|---|---|---|---|---|---|
| birth | | r(55)=0.036, p=0.789 | r(58)=−0.122, p=0.353 | r(56)=−0.138, p=0.303 | r(56)=−0.153, p=0.252 | r(55)=−0.149, p=0.268 | r(53)=−0.137, p=0.319 |
| 1 month | | | r(64)=−0.138, p=0.268 | r(61)=−0.113, p=0.379 | r(62)=−0.113, p=0.294 | r(59)=−0.013, p=0.919 | r(59)=−0.069, p=0.597 |

*Continued*

| **Cognitive flexibility in older preschoolers** | | | | |
|---|---|---|---|---|
| 5 months | | r(72)=−0.002, p=0.984 | r(73)=−0.025, p=0.829 | r(68)=0,064, p=0.601 | r(70)=0.046, p=0.704 |
| 8 months | | | r(72)=0.026, p=0.827 | r(68)=0.098, p=0.421 | r(69)=0.034, p=0.777 |
| 12 months | | | | r(68)=0.091, p=0.456 | r(70)=−0.057, p=0.634 |
| 18 months | | | | | r(65)=−0.103, p=0.406 |
| 24 months | | | | | |

**Appendix 1—table 4.** Results of the correlational analyses between changes in functional connectivity between 5 and 24 months and cognitive flexibility in younger and older preschoolers.

| | Cognitive flexibility | |
|---|---|---|
| ΔFC between 5 and 24 months | Younger preschoolers | Older preschoolers |
| Frontal interhemispheric connectivity | r(12)=−0.368, p=0.196 | r(20)=−0.188, p=0.401 |
| Left fronto-middle FC | r(10)=0.253, p=0.428 | r(17)=0.270, p=0.264 |
| Right fronto-middle FC | r(12)=0.248, p=0.392 | r(20)=0.192, p=0.392 |
| Right frontal-posterior FC | r(12)=0.188, p=0.520 | r(20)=0.164, p=0.465 |

**Appendix 1—table 5.** Characteristics of included and excluded participants and seconds of data included in the analyses at each age.

WD = withdrawn, D = deceased, MV = missed visit, DD = developmental delay, NIRS not undertaken = the participant was assessed but did not want or could not perform the NIRS assessments, FC not undertaken = the participant was assessed with other NIRS task, but not FC, Fussed out = the participant wore the headband and the FC acquisition had started but the participant showed signs of fussiness soon after the start of the acquisition, MP = missing pictures of the headband placement, EM = missing event markers, TI = technical issues during the NIRS testing session. The proportion of children included in the analysis was computed based on the infants with FC data.

| Age | N -% | Not tested | | | | NIRS not undertaken | FC not undertaken | Infants with FC data | Experimental errors | | | | | | Not enough data after pre-processing | Infants included in the analyses | Seconds of data included in the analyses (mean ± SD) | Inclusion rate (from the 204 infants recruited) |
| --- | --- | --- | --- | --- | --- | --- | --- | --- | --- | --- | --- | --- | --- | --- | --- | --- | --- | --- |
| | | WD | D | MV | DD | | | | Fussed out | MP | EM | TI | Headband Placement | Too many channels excluded | | | | |
| 5 months | N | 2 | 1 | 2 | 3 | 10 | 7 | 179 | 16 | 11 | 3 | 3 | 7 | 5 | 47 | 87 | | |
| | % | 0.98 | 0.49 | 0.98 | 1.47 | 4.90 | 3.43 | 87.74 | 8.93 | 5.39 | 1.67 | 1.67 | 5.58 | 2.79 | 26.25 | 48.6 | 382.68±92.78 | 42% |
| 8 months | N | 7 | 1 | 5 | 3 | 18 | 14 | 156 | 7 | 6 | 0 | 1 | 23 | 6 | 60 | 53 | | |
| | % | 3.43 | 0.49 | 2.45 | 1.47 | 8.82 | 6.86 | 76.47 | 4.48 | 3.84 | 0 | 0.64 | 14.7 | 3.84 | 38.46 | 33.97 | 372.93±80.66 | 25% |
| 12 months | N | 9 | 1 | 4 | 3 | 17 | 13 | 157 | 4 | 1 | 0 | 2 | 10 | 2 | 56 | 82 | | |
| | % | 4.41 | 0.49 | 1.96 | 1.47 | 8.33 | 6.37 | 76.96 | 2.54 | 0.63 | 0 | 1.27 | 6.36 | 1.27 | 35.66 | 52.22 | 372.93±80.66 | 40% |
| 18 months | N | 8 | 1 | 15 | 3 | 12 | 5 | 160 | 4 | 0 | 0 | 1 | 16 | 4 | 38 | 97 | | |
| | % | 3.92 | 0.49 | 7.35 | 1.47 | 5.88 | 2.45 | 78.43 | 2.5 | 0 | 0 | 0.62 | 10 | 2.5 | 23.75 | 60.62 | 388.85±81.63 | 47% |
| 24 months | N | 13 | 1 | 29 | 1 | 3 | 4 | 153 | 0 | 2 | 0 | 0 | 6 | 4 | 45 | 96 | | |
| | % | 6.37 | 0.49 | 14.2 | 0.49 | 1.47 | 1.96 | 75 | 0 | 1.30 | 0 | 0 | 3.92 | 2.61 | 29.41 | 62.74 | 399.56±79.86 | 47% |

