## [Editor Report · eLife Assessment]

This **important** study details changes in the brain functional connectivity in a longitudinal cohort of Gambian children assessed outside a lab setup with functional near-infrared spectroscopy (fNIRS) from age 5 to 24 months, in relation to early physical growth and cognitive flexibility capacities at preschool age. Evidence supporting conclusions on the evolution of brain connectivity is **convincing** and highlights a different trajectory compared with populations from high-income countries. However, analyses linking connectivity trajectories with early adverse conditions such as undernutrition and later cognitive development are only partially supported due to insufficient longitudinal data and statistical power. This study will be of significant interest to neuroscientists, psychologists and neuroimaging researchers working on infant development in relation to environmental factors.

---

## [Referee Report · Reviewer #1 (Public review)]

Summary:

This study utilizes fNIRS to investigate the effects of undernutrition on functional connectivity patterns in infants from a rural population in Gambia. fNIRS resting-state data recording spanned ages 5 to 24 months, while growth measures were collected from birth to 24 months. Additionally, executive functioning tasks were administered at 3 or 5 years of age. The results show an increase in left and right frontal-middle and right frontal-posterior connections with age and, contrary to previous findings in high-income countries, a decrease in frontal interhemispheric connectivity. Restricted growth during the first months of life was associated with stronger frontal interhemispheric connectivity and weaker right frontal-posterior connectivity at 24 months of age. Additionally, the study describes some connectivity patterns, including stronger frontal interhemispheric connectivity, which is associated with better cognitive flexibility at preschool age.

Strengths:

- The study analyses longitudinal data from a large cohort (n = 204) of infants living in a rural area of Gambia. This already represents a large sample for most infant studies, and it is impressive, considering it was collected outside the lab in a population that is underrepresented in the literature. The research question regarding the effect of early nutritional deficiency on brain development is highly relevant and may highlight the importance of early interventions. The study may also encourage further research on different underrepresented infant populations (i.e., infants not residing in Western high-income countries) or in settings where fMRI is not feasible.

- The preprocessing and analysis steps are carefully described, which is very welcome in the fNIRS field, where well-defined standards for preprocessing and analysis are still lacking.

Weaknesses:

- The study provides a solid description of the functional connectivity changes in the first two years of life at the group level and investigates how restricted growth influences connectivity patterns at 24 months. However, it does not explore the links between adverse situations and developmental trajectories for functional connectivity. Given the longitudinal nature of the dataset, future work should expand the analysis using more sophisticated tools to link undernutrition to specific developmental trajectories in functional connectivity, and eventually incorporate additional data to increase statistical power.

- Connectivity was assessed in 6 big ROIs to reduce variability due to head size and optode placement. Nevertheless, this also implies a significant reduction in spatial resolution. Individual digitalisation and co-registration of the optodes to a head model, followed by image reconstruction, could provide better spatial resolution. This is not a weakness specific to this study but rather a limitation common to most fNIRS studies, which typically analyse data at the channel level since digitalisation and co-registration can be challenging, especially in complex setups like this. The authors made an important effort to identify subjects with major optode displacement; however, future work might use tools to digitally record the positions of optodes and head markers.

---

## [Referee Report · Reviewer #2 (Public review)]

Strengths:

The article addresses a topic of significant importance, focusing on early life growth faltering in low-income countries-a key marker of undernutrition-and its impact on brain functional connectivity (FC) and cognitive development. The study's strengths include the laborious data collection process, as well as the rigorous data preprocessing methods employed to ensure high data quality. The use of cutting-edge preprocessing techniques further enhances the reliability and validity of the findings, making this a valuable contribution to the field of developmental neuroscience and global health.

Weaknesses:

The study lacks specificity in identifying which specific brain networks are affected by growth faltering, as the current exploratory analyses mainly provide an overall conclusion that infant brain network development is impacted without pinpointing the precise neural mechanisms or networks involved.

---

## [Referee Report · Reviewer #3 (Public review)]

Summary

This study aimed to investigate whether the development of functional connectivity (FC) is modulated by early physical growth, and whether these might impact cognitive development in childhood. This question was investigated by studying a large group of infants (N=204) assessed in Gambia with fNIRS at 5 visits between 5 and 24 months of age. Given the complexity of data acquisition at these ages and following data processing, data could be analyzed for 53 to 97 infants per age group. FC was analyzed considering 6 ensembles of brain regions and thus 21 types of connections. Results suggested that: (i) compared to previously studied groups, this group of Gambian infants have different FC trajectory, in particular with a change in frontal inter-hemispheric FC with age from positive to null values; (ii) early physical growth, measured through weight-for-length z-scores from birth onwards, is associated with FC at 24 months. Some relationships were further observed between FC during the first two years and cognitive flexibility, in different ways between 4- and 5-year-old preschoolers, but results did not survive corrections for multiple comparisons.

Strengths

The question investigated in this article is important for understanding the role of early growth and undernutrition on brain and behavioral development in infants and children. The longitudinal approach considered is highly relevant to investigate neurodevelopmental trajectories. Furthermore, this study targets a little studied population from a low-/middle-income country, which was made possible by the use of fNIRS outside the lab environment. The collected dataset is thus impressive and it opens up a wide range of analytical possibilities.

Weaknesses

Data analyses were constrained by the limited number of children with longitudinal data on NIRS functional connectivity. Applying advanced statistical modeling approaches such as structural equation modelling would provide further insights on neurodevelopmental trajectories and relationships with early growth and later cognitive development.

---

## [Author Response]

The following is the authors’ response to the previous reviews

**Public Reviews:**

**Reviewer #1 (Public review):**
Summary:This study utilises fNIRS to investigate the effects of undernutrition on functional connectivity patterns in infants from a rural population in Gambia. fNIRS resting-state data recording spanned ages 5 to 24 months, while growth measures were collected from birth to 24 months. Additionally, executive functioning tasks were administered at 3 or 5 years of age. The results show an increase in left and right frontal-middle and right frontal-posterior connections with age and, contrary to previous findings in high-income countries, a decrease in frontal interhemispheric connectivity. Restricted growth during the first months of life was associated with stronger frontal interhemispheric connectivity and weaker right frontal-posterior connectivity at 24 months of age. Additionally, the study describes some connectivity patterns, including stronger frontal interhemispheric connectivity, which is associated with better cognitive flexibility at preschool age.Strengths:The study analyses longitudinal data from a large cohort (n = 204) of infants living in a rural area of Gambia. This already represents a large sample for most infant studies, and it is impressive, considering it was collected outside the lab in a population that is underrepresented in the literature. The research question regarding the effect of early nutritional deficiency on brain development is highly relevant and may highlight the importance of early interventions. The study may also encourage further research on different underrepresented infant populations (i.e., infants not residing in Western high-income countries) or in settings where fMRI is not feasible.The preprocessing and analysis steps are carefully described, which is very welcome in the fNIRS field, where well-defined standards for preprocessing and analysis are still lacking.

We thank the reviewer for highlighting the strengths of this work.

Weaknesses:While the study provides a solid description of the functional connectivity changes in the first two years of life at the group level and investigates how restricted growth influences connectivity patterns at 24 months, it does not explore the links between adverse situations and developmental trajectories for functional connectivity. Considering the longitudinal nature of the dataset, it would have been interesting to apply more sophisticated analytical tools to link undernutrition to specific developmental trajectories in functional connectivity. The authors mention that they lack the statistical power to separate infants into groups according to their growing profiles. However, I wonder if this aspect could not have been better explored using other modelling strategies and dimensional reduction techniques. I can think about methods such as partial least squares correlation, with age included as a numerical variable and measures of undernutrition.

We agree with the reviewer that this complex and rich longitudinal dataset would benefit from more sophisticated analytical approaches to characterise developmental trajectories in functional connectivity and to more directly link them to measures of undernutrition. However, conducting such analyses would require substantial additional methodological development, model validation, and careful interpretation, which fall beyond the scope and timeline of the present manuscript. Our aim here was to provide a clear and robust characterisation of functional connectivity changes during the first two years of life and to examine associations with growth outcomes at a specific developmental stage, while ensuring methodological transparency and statistical reliability. Importantly, these more advanced trajectory-based analyses are currently being pursued in the final phase of the BRIGHT project (BRIGHT IMPACT), in collaboration with expert statisticians and data scientists. This ongoing work aims specifically to leverage the longitudinal richness of the dataset to model developmental trajectories and their associations with early-life adversity and nutritional factors. We therefore see the present study as an important foundation for these forthcoming analyses.

Connectivity was assessed in 6 big ROIs. While the authors justify this choice to reduce variability due to head size and optodes placement, this also implies a significant reduction in spatial resolution. Individual digitalisation and co-registration of the optodes to the head model, followed by image reconstruction, could have provided better spatial resolution. This is not a weakness specific to this study but rather a limitation common to most fNIRS studies, which typically analyse data at the channel level since digitalisation and co-registration can be challenging, especially in complex setups like this. However, the BRIGHT project has demonstrated that it is possible and that differences in placement affect activation patterns, which become more localised when data is co-registered at the subject level (Collins-Jones et al., 2021). Could the co-registration of individual data have increased sensitivity, particularly given that longitudinal effects are being investigated?

We agree with the reviewer that the fNIRS community should work toward more precise methods for spatial registration of optodes, not only at the group level but also at the subject level, in order to make more precise inferences about the locations of activations. However, we followed a very thorough offline procedure to model headgear placement based on each participant’s photographs, which we believe complements the coregistration work performed by Collins-Jones in 2021. As reported in the fNIRS data acquisition section “Infants were excluded from further analysis if the band was excessively high over the front above the eyebrows” (line 409, methods section). Moreover channels displacement was measured from the photos, and if it was “equal or greater than 1.6 cm were renumbered, so that each channel was shifted either backward or forward one full channel location in space” (line 413, methods section). While these practices are thoroughly followed in the BRIGHT project, we are aware that they are not part of the standard procedure in many infant fNIRS studies. We hope that this work provides guidance for other researchers on how to coregister infant fNIRS data.

Considering the spatial resolution of fNIRS, which is on the order of centimetres, and the thorough procedure combining fNIRS–MRI coregistration with channel displacement assessment based on photographs, we do not think that individual-level coregistration would have significantly increased the sensitivity of the results.

I believe that a further discussion in the manuscript on the application of global signal regression and its effects could have been beneficial for future research and for readers to better understand the negative correlations described in the results. Since systemic physiological changes affect HbO/HbR concentrations, resulting in an overestimation of functional connectivity, regressing the global signal before connectivity computation is a common strategy in fNIRS and fMRI studies. However, the recommendation for this step remains controversial, likely depending on the case (Murphy & Fox, 2017). I understand that different reasons justify its application in the current study. In addition to systemic physiological changes originating from brain tissue, fNIRS recordings are contaminated by changes occurring in superficial layers (i.e., the scalp and skull). While having short-distance channels could have helped to quantify extracerebral changes, challenges exist in using them in infant populations, especially in a longitudinal study such as the one presented here. The optimal source-detector distance that minimises sensitivity to changes originating from the brain would increase with head size, and very young participants would require significantly shorter source-detector distances (Brigadoi & Cooper, 2015). Thus, having them would have been challenging. Under these circumstances (i.e., lack of short channels and external physiological measures), and considering that the amount the signal is affected by physiological noise (either coming from the brain or superficial tissue) might change through development, the choice of applying global signal regression is justified. Nevertheless, since the method introduces negative correlations in the data by forcing connectivity to average to zero, I believe a further discussion of these points would have enriched the interpretation of the results.

We added a paragraph discussing the choice of using GSR in our pipeline in the discussion of the manuscript as follows: “Importantly, these results remained significant even without GSR, indicating that our findings are not solely driven by preprocessing choices. While the use of GSR in FC studies remains debated (Murphy & Fox, 2017), in the absence of short channels (which are difficult to use reliably with infants (Emberson et al., 2016)) and external physiological measures, applying GSR represented the most appropriate preprocessing option. In fact, failure to correct for systemic physiological fluctuations can, in fact, lead to artificially elevated connectivity estimates in fNIRS data (Abdalmalak et al., 2022)” (line 250, discussion section).

**Reviewer #2 (Public review):**
Strengths:The article addresses a topic of significant importance, focusing on early life growth faltering in low-income countries-a key marker of undernutrition-and its impact on brain functional connectivity (FC) and cognitive development. The study's strengths include the laborious data collection process, as well as the rigorous data preprocessing methods employed to ensure high data quality. The use of cutting-edge preprocessing techniques further enhances the reliability and validity of the findings, making this a valuable contribution to the field of developmental neuroscience and global health.

We thank the reviewer for highlighting the strengths of this work.

Weaknesses:The study fails to fully leverage its longitudinal design to explore neurodevelopmental changes or trajectories, as highlighted by all three reviewers. The revised manuscript still primarily focuses on FC values at a single age stage (i.e., 24 months) rather than utilizing the longitudinal data to investigate how FC evolves over time or predicts cognitive development. Although the authors acknowledge that analyzing changes in FC (ΔFC) would reduce degrees of freedom (to ~30) and risk interpretability, they do not report or discuss these results, even as exploratory findings.

As suggested, we added the table reporting the results of the associations between changes in functional connectivity (DFC) between 5 and 24 months and cognitive flexibility in the supplementary materials (Table SI3). We additionally explored the relationship between changes in growth and cognitive flexibility as suggested by Reviewer #3 and we reported these additional analyses in the text as follows: “We also explored whether changes in growth and changes in functional connectivity between 5 and 24 months were associated with cognitive flexibility at preschool age, but we did not find any significant association (Table SI3 and Table SI4).” (line 213, results section).

Furthermore, the study lacks specificity in identifying which specific brain networks are affected by growth faltering, as the current exploratory analyses mainly provide an overall conclusion that infant brain network development is impacted without pinpointing the precise neural mechanisms or networks involved.

We added this limitation in the discussion as follows: “While the impact of undernutrition on brain development has been documented in LMICs (46), herein, we provided empirical evidence that growth faltering specifically in infants younger than five months of age impacts observable development of functional brain networks in the second year of life. Future studies may be needed to pinpoint which specific brain networks are impacted” (line 279, discussion section).

**Reviewer #3 (Public review):**
SummaryThis study aimed to investigate whether the development of functional connectivity (FC) is modulated by early physical growth, and whether these might impact cognitive development in childhood. This question was investigated by studying a large group of infants (N=204) assessed in Gambia with fNIRS at 5 visits between 5 and 24 months of age. Given the complexity of data acquisition at these ages and following data processing, data could be analyzed for 53 to 97 infants per age group. FC was analyzed considering 6 ensembles of brain regions and thus 21 types of connections. Results suggested that: (i) compared to previously studied groups, this group of Gambian infants have different FC trajectory, in particular with a change in frontal inter-hemispheric FC with age from positive to null values; (ii) early physical growth, measured through weight-for-length z-scores from birth on, is associated with FC at 24 months. Some relationships were further observed between FC during the first two years and cognitive flexibility, in different ways between 4- and 5-year-old preschoolers, but results did not survive corrections for multiple comparisons.StrengthsThe question investigated in this article is important for understanding the role of early growth and undernutrition on brain and behavioral development in infants and children. The longitudinal approach considered is highly relevant to investigate neurodevelopmental trajectories. Furthermore, this study targets a little studied population from a low-/middle-income country, which was made possible by the use of fNIRS outside the lab environment. The collected dataset is thus impressive and it opens up a wide range of analytical possibilities.

We thank the reviewer for highlighting the strengths of this work.

WeaknessesData analyses were constrained by the limited number of children with longitudinal data on NIRS functional connectivity. Nevertheless, considering more advanced statistical modelling approaches would be relevant to further explore neurodevelopmental trajectories as well as relationships with early growth and later cognitive development.

While in this study we selected specific FC and outcome variables based on our hypothesis, the final phase of the BRIGHT project, known as BRIGHT IMPACT, aims to apply advanced statistical models to integrate a range of project variables into a single comprehensive analysis. We have acknowledged this in the discussion as follows: “Applying more advanced statistical modelling methods and structural equation modelling analyses may provide greater insight with further investigations in contexts of adversity and, in turn, establish which outcomes are predicted by FC” (line 309, discussion section).

The abstract and end of the discussion should make it clearer that the associations between FC and cognitive flexibility are results that need to be confirmed, insofar as they did not survive correction for multiple comparisons.

We have acknowledged this in the abstract as follows: “Our results highlight the measurable effects that poor growth in early infancy has on brain development and the possible subsequent impact on pre-school age cognitive development, underscoring the need for early life interventions throughout global settings of adversity”.

We have acknowledged this in the discussion as follows: “While our results are consistent with previous studies, we acknowledge that the significant associations between early FC and later cognitive flexibility do not withstand multiple comparisons. Therefore, we encourage future studies that may replicate these findings with a larger sample” (line 300, discussion section).

**Recommendations for the authors:**

**Reviewer #1 (Recommendations for the authors):**
(1) In Figure 1 B and C the authors should indicate that the results refer to HbO.

We have added the suggested specification in the caption of the figure as suggested.

(2) Figure SI2. Please indicate in the caption that these are the results when pre-processing did not include global signal regression.

We have added the suggested specification in the caption of the figure as suggested.

**Reviewer #3 (Recommendations for the authors):**
(1) The sentence l529-531 ("To investigate whether FC early in life predicted...") should be more explicit as it is not clear which of the two variables is regressed by the other: is it the measure of cognitive flexibility that is regressed by FC, as the hypothesis suggests? Were other variables considered in the regression model? (For linear regression with only one "prediction" variable, the square root of the coefficient of determination 𝑅2 is equal to the correlation between the two variables.)

Yes, it is the measure of cognitive flexibility that is regressed by FC. We have rephrased it in the text as follows: “we regressed later cognitive flexibility against FC that showed a significant change across the first two years of life”. There were no other variables in the regression model.

(2) A summary table of the statistical results for FC-cognitive flexibility associations should be included as for other analyses, in addition to Figure 3B.

We added a table of the results for the association between FC and cognitive flexibility in the supplementary materials (Table SI2, page 10), matching the same colours of Table 2. We referenced the table in the text in the main manuscript (line 211, result section).

(3) Figure 3B: The legend should precise that these results did not survive corrections for multiple comparisons.

We have specified this in the legend of Figure 3 as suggested.

(4) For the young pre-schooler group, it seems that the age is around 4 years (age mean +/- SD=47.96 +/- 2.77 months) and not 3 years as indicated at several places in the manuscript.

We found only once instance in which we erroneously said that the younger preschoolers were around 3 years. We replaced “Gambian infants from BRIGHT were cross-sectionally assessed at the age of 3 or 5 years for cognitive flexibility” with Gambian infants from BRIGHT were cross-sectionally assessed between the age of 3 and 5 years for cognitive flexibility (line 489, method section).

(5) The authors use the term "intra-hemispheric" connections for the ones within each of the 6 sections. This might be misleading since fronto-posterior connections are also intra-hemispheric ones. Specifying "short-range" or "within-section" connections might be clearer.

As suggested by the reviewer, we replaced “intra-hemispheric” with “intra-hemispheric within section” where appropriate through the whole manuscript.

(6) Abstract: what is the justification for using the term "optimal" for describing developmental trajectories of FC?

The term “optimal” refers to knowledge about typical developmental trajectories, coming especially from fMRI studies, as mentioned in the introduction: “Based on data from fMRI, current models hypothesize that FC patterns mature throughout early development (23–27), where in typically developing brains, adult-like networks emerge over the first years of life as long-range functional connections between pre-frontal, parietal, temporal, and occipital regions become stronger and more selective (28–31). [...]. Importantly, normative developmental patterns may be disrupted and even reversed in clinical conditions that impact development; e.g., increased short-range and reduced long-range FC have been observed in preterm infants (36) and in children with autism spectrum disorder (37, 38)” (line 93-106, introduction).

(7) The confidence interval should be added in Figure SI3.

As suggested, confidence intervals have been added in Figure SI3.

(8) Other scatterplot examples of associations might be added as supplementary information.

As suggested, we added several additional scatterplots to Figure SI3 (with confidence intervals as noted in the comment above) to show other associations between changes in growth and FC at 24 months.

(9) Figure SI6: % in x-axis is still indicated.

We apology for the oversight, all the percentage signs have now been removed from the x-axis tick labels.

(10) The authors might show the (even not significant) results of the associations between changes in growth and cognitive flexibility in supplementary information.

As suggested, we added the table reporting the results of the associations between changes in growth (DWLZ) and cognitive flexibility in the supplementary materials (Table SI3). We additionally explored the relationship between changes in functional connectivity and cognitive flexibility as suggested by Reviewer #2 and we reported these additional analyses in the text as follows: “We also explored whether changes in growth and changes in functional connectivity between 5 and 24 months were associated with cognitive flexibility at preschool age, but we did not find any significant association (Table SI3 and Table SI4).” (line 213, results section).